# Rapid detection and recognition of whole brain activity in a freely behaving *Caenorhabditis elegans*

**Yuxiang Wu**[1,2], **Shang Wu**[1,2], **Xin Wang**[3], **Chengtian Lang**[4], **Quanshi Zhang**[3], **Quan Wen**[1,2]*, **Tianqi Xu**[1,2]*

**1** Chinese Academy of Sciences Key Laboratory of Brain Function and Diseases, Division of Life Sciences and Medicine, University of Science and Technology of China, Hefei, China, **2** Hefei National Laboratory for Physical Sciences at the Microscale, Center for Integrative Imaging, University of Science and Technology of China, Hefei, China, **3** John Hopcroft Center for Computer Science, School of electronic information and electrical engineering, Shanghai Jiao Tong University, Shanghai, China, **4** Department of Electronic Engineering and Information Science, University of Science and Technology of China, Hefei, China

* qwen@ustc.edu.cn (QW); xutq@ustc.edu.cn (TX)

## Abstract

Advanced volumetric imaging methods and genetically encoded activity indicators have permitted a comprehensive characterization of whole brain activity at single neuron resolution in *Caenorhabditis elegans*. The constant motion and deformation of the nematode nervous system, however, impose a great challenge for consistent identification of densely packed neurons in a behaving animal. Here, we propose a cascade solution for long-term and rapid recognition of head ganglion neurons in a freely moving *C. elegans*. First, potential neuronal regions from a stack of fluorescence images are detected by a deep learning algorithm. Second, 2-dimensional neuronal regions are fused into 3-dimensional neuron entities. Third, by exploiting the neuronal density distribution surrounding a neuron and relative positional information between neurons, a multi-class artificial neural network transforms engineered neuronal feature vectors into digital neuronal identities. With a small number of training samples, our bottom-up approach is able to process each volume—1024 × 1024 × 18 in voxels —in less than 1 second and achieves an accuracy of 91% in neuronal detection and above 80% in neuronal tracking over a long video recording. Our work represents a step towards rapid and fully automated algorithms for decoding whole brain activity underlying naturalistic behaviors.

## Author summary

An important question in neuroscience is to understand the relationship between brain dynamics and naturalistic behaviors when an animal is freely exploring its environment. In the last decade, it has become possible to genetically engineer animals whose neurons produce fluorescence reporters that change their brightness in response to brain activity. In small animals such as the nematode *C. elegans*, we can now record the fluorescence changes in and thereby infer neural activity from most neurons in the head of a worm,

**Funding:** This work was supported by Major International (Regional) Joint Research Project (32020103007) to Q.W., by China Postdoctoral Research Foundation (project number 2020M682016) to T.X., partially by National Key R \&D Program of China (2021ZD0111602) to Q.Z., and by the Strategic Priority Research Program of the Chinese Academy of Sciences (Pilot study, grant XDPB10, Grant No. XDB39000000) to Y.W., S.W., Q.W., and T.X. The funders had no role in study design, data collection and analysis, decision to publish, or preparation of the manuscript.

**Competing interests:** The authors have declared that no competing interests exist.

when the animal is freely moving. These neurons are densely packed in a small volume. Since the brain and body are moving and its shape undergoes significant deformation, a human expert, even after long hours of inspection, may still have difficulty to tell where the neurons are and who they are.

We sought to develop an automatic method for rapidly detecting and tracking most of these neurons in a moving animal. To do this, we asked a human expert to annotate all head neurons—their locations and digital identities—across a small number of volumes. Then, we trained a computer to learn the locations and digital identities of these neurons across different imaging volumes. Our machine inference method is fast and accurate. While it takes a human expert several hours to complete a sequence of volumes, a machine can finish the task in a few minutes. We hope our method provides a better and more efficient engine for extracting knowledge from whole brain imaging datasets and animal behaviors.

This is a *PLOS Computational Biology* Methods paper.

## Introduction

Characterizing whole-brain activity is crucial for opening the black box of a biological neural network. Recent technological advances in genetically encoded calcium and voltage indicators, volumetric imaging, and behavioral tracking make it possible to simultaneously record the activity of a large fraction of neurons in an unrestrained animal [1–9], opening the door to an exploration of brain dynamics and naturalistic behavior at multiple spatiotemporal scales.

Over time, constant deformation of a nematode brain, which causes large displacement and distortion of neurons, has been a major hindrance to rapid and accurate characterization of neural activity patterns. Furthermore, highly similar textures and shapes between neurons within an animal, as well as highly variable distributions of neurons across animals, make detection and identification of neurons an extraordinarily challenging problem.

Extracting whole-brain activity from an unrestrained *C. elegans* requires answering two questions: *where are the neurons* and *who are they*? Conventional image processing methods and deep-learning-based methods have been proposed to solve both problems [10–13]. To answer the first question, traditional methods use explicit computational models to localize neuronal regions, for example, by computing the 3D Hessian matrix in the pixel space [11] and by implementing the watershed algorithm for regional segmentation [11, 13]. Deep-learning-based methods, such as 3D-UNet [13, 14], train feed-forward neural network models to find candidate neuronal regions. With a sufficient number of human-annotated supervised examples, a neural network can effectively and accurately perform the neuron detection task.

Sequence-dependent and sequence-independent methods have been proposed to address the second question, namely to assign a digital identity to each neuron in every imaging volume. For example, the recently developed 3DeeCellTracker [13] is a sequence-dependent method, by which a tracking or registration algorithm maps the correspondence between two sets of neurons in adjacent volumes. 3DeeCellTracker would work well for semi-immobilized animals, when cell displacements are relatively small and spatiotemporal continuity of imaging

 

volumes approximately holds; but it is prone to error accumulation when cell displacements are large, rapid and not easily predictable. By adapting the single-particle-tracking framework, Lagache et al. [15] proposed a method called Elastic Motion Correction and Concatenation ($EMC^2$) to deal with the tracking gap: neurons may appear and disappear in consecutive time frames depending on whether calcium florescent signals can be detected. $EMC^2$ was successfully applied to extract neural activity in the highly deformable *Hydra*.

Sequence-independent methods allocate neuron identities by ignoring the temporal order of imaging volumes. In an earlier work, Nguyen et al. [11] carefully built a 300-dimensional cluster vector for each neuron in every volume by mapping its correspondence to a point set in each of the 300 reference volumes, and then allocated digital identities all at once by a hierarchical clustering algorithm. In a different approach, an algorithm aims to find optimal neuron assignment by maximizing the intrinsic similarity between a point-set in a testing volume and that in a different imaging volume. The intrinsic similarity could be cast in a deep neural network model, called fDNC [12]. In fDNC, large synthetic point clouds across spacetime, which mimic the movements of *C. elegans* head neurons, were used to train a transformer neural network. During inference, the transformer rapidly encodes each neuron's feature vector, computes pairwise similarity with those in a reference volume, and allocates each neuron's digital identity.

Cell matching can also be carried out between an imaging volume and a standard annotated atlas (e.g., OpenWorm [16] or NeuroPAL [17]) with known cell-type identities (see Digital ID, cell-type ID, and human annotation for a definition). Neurons or muscle cells in fixed and straightened worms can be recognized and mapped to the atlas using registration methods [18] and more recently, probabilistic models that take into account variability of cell positions [19–21] and take advantage of color information [17] to improve the recognition accuracy [20, 21]. In this paper, we restrict ourselves to matching neuronal digital identities within an animal; matching cell-type identities across animals is beyond the scope of the present work.

With recent development in machine vision and deep learning, we are asking for fast, efficient, and user-friendly algorithms that take, with minimum human supervision and correction, raw volumetric imaging data as inputs and extract neural activity traces from all recorded neurons. Unlike in immobilized animals where cell positions are invariant and spatiotemporal segmentation could be implemented at once [22–24], 3D neuron detection and segmentation in freely behaving *C. elegans* is carried out in every imaging volume independently and has become the speed bottleneck [11, 13]. For example, 3DeeCellTracker requires specialized processing steps such as straightening a worm [11, 13] to facilitate neuron mapping and error corrections. This procedure, together with 3D registration and segmentation, is the most time-consuming step. To identify neurons across volumes, both fDNC and 3DeeCellTracker [12, 13] require enormous training samples (e.g., large computer-generated synthetic datasets). It remains to be evaluated whether neural networks trained by data augmentation possess sufficient generalization power when the algorithms are applied to datasets collected from a different transgenic animal and by a different imaging system.

Here, we propose a streamlined approach, called CeNDeR (**C**. *elegans* **N**euron **De**tection and **R**ecognition), to extract whole brain neural activity in a freely behaving *C. elegans*. As in previous works [1, 2], we combine a spinning disk confocal microscope and a customized tracking system to record neural activity in the head ganglion while the animal is moving. We use nuclei-localized red fluorescent protein (mNeptune) as a reference channel to localize and recognize neurons. The CeNDeR system processes an imaging volume in the following 4 steps. 1) An automatic pre-processing procedure segments the head region and builds a *C. elegans* coordinate system. 2) A multi-field detection (MFD) neural network finds, with high confidence, a set of neuronal regions in each frame. 3) By treating regional alignment across adjacent frames as a maximum bipartite matching problem, we use the Hungarian method to

merge candidate regions into a neuron. 4) A multi-class recognition neural network assigns a digital ID to every neuron in a random volume using two input feature vectors, designed to extract neuronal density and relative positional information surrounding a neuron. When taking into account speed-accuracy trade-off, our approach complements recent methods such as 3DeeCellTracker and fDNC. With a small number ($\sim 30$) of human-annotated training examples, CeNDeR can process each volume in less than a second and achieve an accuracy of 88.25% when tracking head ganglion neurons within an animal.

## Methods

The CeNDeR pipeline (Fig 1) takes a volume of size $W \times H \times Z$ as an input (Fig 1A), and outputs neuron objects including each neuron's digital identity, location, and shape. Our approach comprises the following four steps. First, we use a combination of image processing algorithms to extract head and body regions and build the *C. elegans* coordinate system for every volume (Fig 1B). Second, we identify local pixel intensity maxima in a frame, which will be used as anchor points to build multi-field inputs. A trained neural network takes the inputs and predicts the score of each region as well as its position and shape. By removing overlaps between regions, the algorithm then finds a list of neuronal regions in every frame $\{R_1, R_2, R_3, \ldots\}$ (Fig 1C). Third, $\{R_k\}$ are appropriately merged into a set of neuron objects $\{\mathcal{N}_i \mid i \in \{1, 2, 3, \ldots\}\}$ by the X*NeuroAlignment* algorithm (Fig 1D). Finally, we engineer two feature vectors to characterize each neuron's relationship with its surroundings. The concatenated

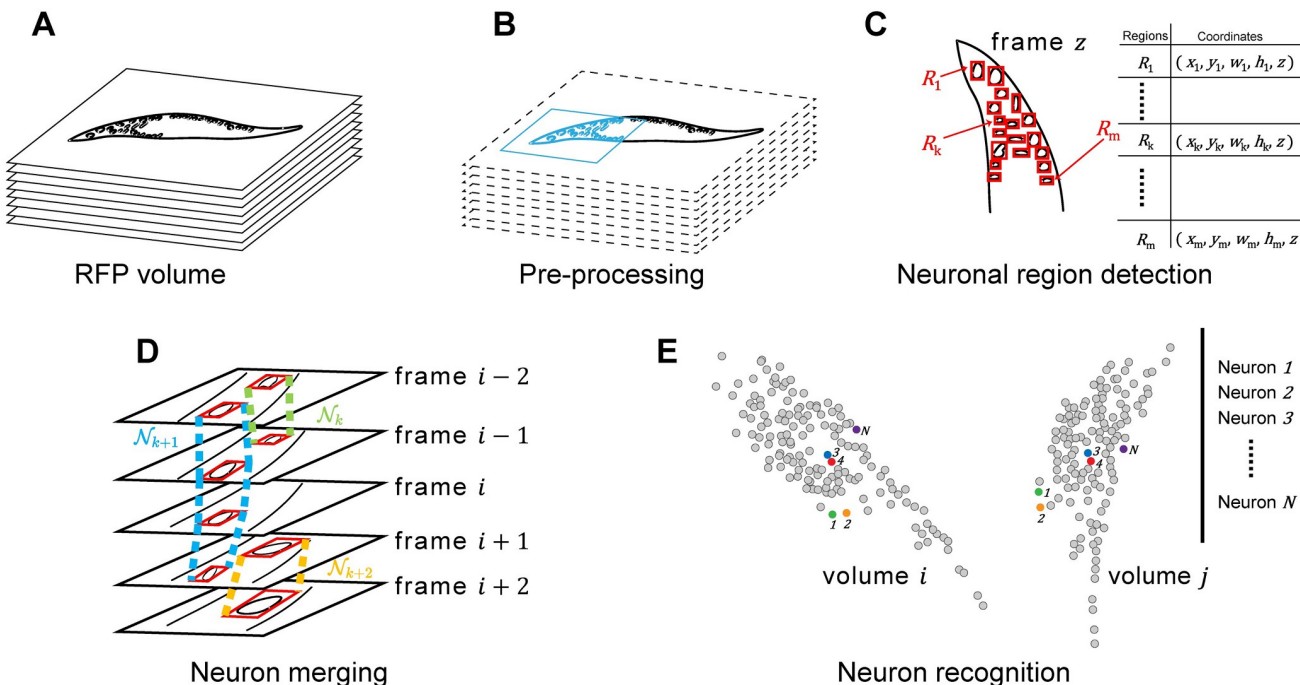

**Fig 1. The CeNDeR pipeline.** **(A)** A red channel (RFP) imaging volume consists of a stack of fluorescence images ($1024 \times 1024 \times 18$, see Section Imaging setup for further details). **(B)** An Automatic Pre-Processing (APP) algorithm crops the head region (blue rectangle) and defines the *C. elegans* coordinate system. **(C)** A Multi-Field Detection (MFD) algorithm generates anchor boxes of different sizes centered at local maxima of pixel intensities. An ANN refines the shapes and locations of anchor boxes and proposes neuronal regions $R$ (red rectangle) in every frame, each of which is represented by a quintuple $(x, y, w, h, z)$. **(D)** A 3D merging procedure views regional alignment across adjacent frames as a maximum bipartite matching problem, solved by the Hungarian method. $\mathcal{N}_k$, $\mathcal{N}_{k+1}$ and $\mathcal{N}_{k+2}$ illustrate how the procedure merges neuronal regions into three neuron objects. **(E)** A multi-class recognition neural network takes designed feature vectors and assigns an digital identity to every neuron in a volume. Neurons sharing the same identity are represented by the same color. $N$ is the number of neurons to be recognized.

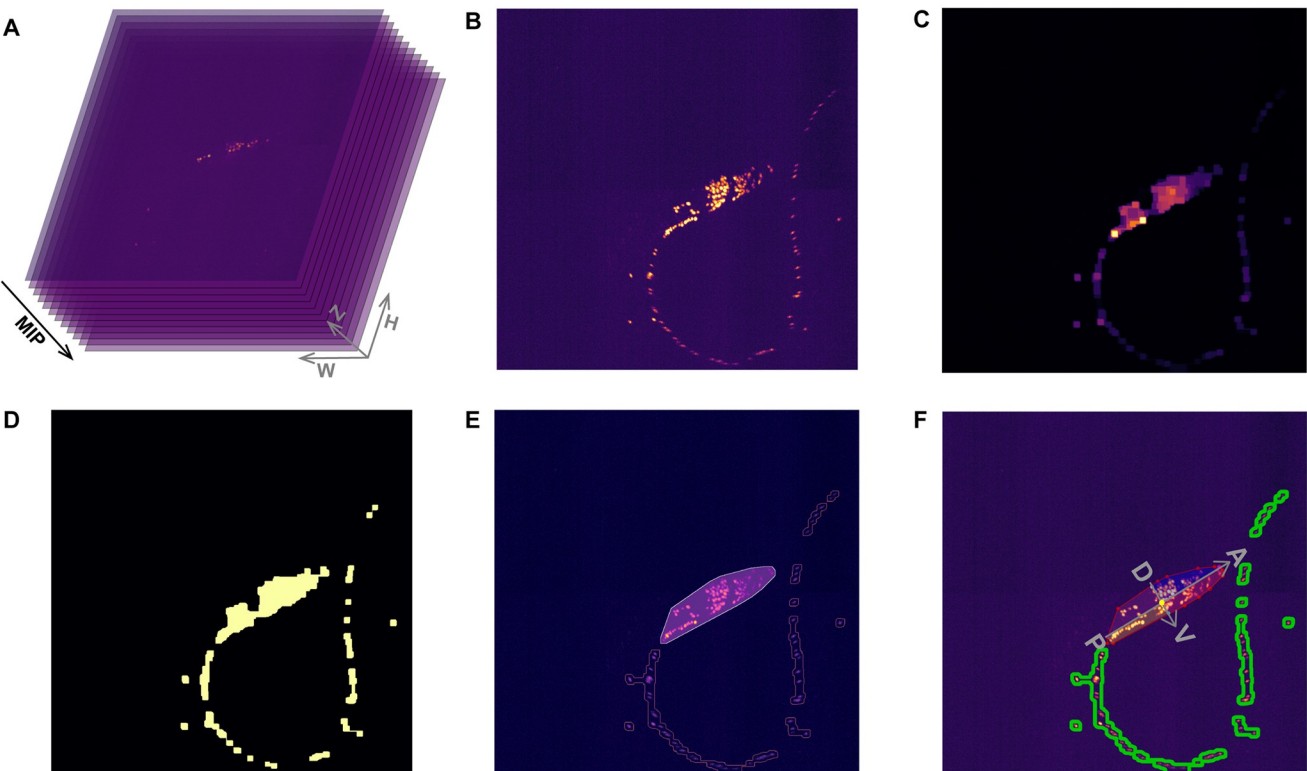

**Fig 2. Automatic Pre-Processing (APP). (A)** A RFP imaging volume. **(B)** Maximum Intensity Projection (MIP) of the imaging volume. Brighter pixels represent fluorescence signals. **(C)** Noises are reduced by median blur filters; a closing operation connects neighboring pixels, from which the worm head region will be extracted. **(D)** A dynamic threshold algorithm converts (C) into a binary image. **(E)** A contour finding algorithm identifies a series of separate regions. The largest area (yellow contour with a purple mask) is the head region; the rest are body regions (pink contour). **(F)** A *C. elegans* coordinate system is built for the head region. The yellow point represents the center of mass. A: anterior, P: posterior, V: ventral and D: dorsal.

features are transformed by a multi-class feed-forward recognition network, trained by a specific loss function, into a set of digital identities $i$, $i \in \{1, \ldots, N\}$, where $N$ is the total number of neuron IDs to be recognized and unified across imaging volumes (Fig 1E).

## Stage 1: Pre-processing

We designed an Automatic Pre-Processing (APP) algorithm to extract a head region and use it to build a *C. elegans* coordinate system (Fig 2). We take the maximum intensity projection image $I_{MIP}$ of each volume (Fig 2B), denoise the image with median blur (Fig 2C), and perform the morphological closing operation—dilation followed by erosion—to find connected regions. To better construct a fully connected head region, we let the size of the dilation filter be slightly bigger than the erosion filter. $I_{MIP}$ is converted into a binary image using the dynamic threshold algorithm (Fig 2D), and the contour tracing algorithm [25] is conducted to find contour points $\{C_{head}\}$ of the head, namely the region with the maximal area (Fig 2E). To build the *C. elegans* coordinate system, we calculate the *anterior–posterior* axis and *ventral–dorsal* axis intrinsic to a worm (see Algorithm 1). Later, all neuronal positions will be transformed into their intrinsic coordinates (see Section Stage 4: Neuron recognition). Note that Algorithm 1 relies on the observation that *C. elegans* crawls on its side, and it would fail to build an accurate coordinate system in extreme scenarios: (1) the head is twisting itself; (2) the head and body are touching

each other during an $\Omega$ turn. These hard examples require more sophisticated algorithms and human corrections, and are beyond the scope of the current paper.

**Algorithm 1** An algorithm to build the *C. elegans* coordinate system

```
Require: the head region I_head in the maximum intensity projection
image I_MIP. Contour of the head {C_head}, the largest connected component
in a binary image.
Ensure: center of mass O, anterior endpoint P_a, posterior endpoint P_p,
ventral endpoint P_v and dorsal endpoint P_d.
1: We define a point of interest whose pixel value ≥ Ī_head + σ(I_head),namely
the sum of standard deviation and mean of I_head. These selected points
are used to compute moments and O.
2: M_00 = ∑_{x,y} I_head(x, y)
3: M_10 = ∑_{x,y} x · I_head(x, y), M_01 = ∑_{x,y} y · I_head(x, y)
4: O ← (M_10/M_00, M_01/M_00)
5: O is the origin. The longest line segment passing O and intersect-
ing {C_head} at (P_a, P_p) defines the anterior-posterior axis.
6: The line segment perpendicular to the anterior-posterior axis
defines the ventral-dorsal axis. It passes O and intersects {C_head} at
(P_v, P_d).
7: The ventral-dorsal axis divides the head into two regions; the
region containing a greater number of points of interest defines the
anterior direction.
8: The anterior-posterior axis divides the head into two regions; the
region containing a greater number of points of interest defines the
ventral direction.
```

## Stage 2: Neuronal region detection

In the second stage, we apply a deep learning algorithm to detect, frame by frame, all regions $\{R_k\}$ that belong to neurons. Each neuronal region is a rectangular bounding box, represented by a quintuple: $R_k = (x, y, w, h, z)$, where $x$ and $y$ are upper left coordinates of the rectangle; $z$ is the frame index of a single volume; $w$ and $h$ are width and height of the rectangle respectively.

Instead of examining each pixel in a frame, the APP algorithm from stage 1 has reduced the searching area to the head region (Fig 3A), much smaller than the size of an entire image. To accelerate neuronal detection, we first identify local pixel intensity maxima in the head region (Fig 3B). These local peaks $\{P_k\}$ likely belong to potential neuronal regions. The center and the size of a candidate region, however, remain unknown. Next, a trained Artificial Neural Network (ANN) is used to obtain these information. Details of the training, validation and test set are provided in the ANN training section. Inputs to the ANN include multiple rectangles centered at $\{P_k\}$, called anchor boxes (Fig 3C) and represented by $A = (x_A, y_A, w_A, h_A, z_A)$, where $x_A$ and $y_A$ are upper left coordinates of an box; $w_A$ and $h_A$ are width and height respectively; $z_A$ is the frame index. Multiple boxes of different sizes are used to cover neurons of various shapes. Multi-field images (Fig 3D Up), which are concentric with the anchor box, provide additional local and context information to the neural network. The ANN predicts a score $S$ for each region, as well as the regional position and size, formulated as corrections, $(\Delta\hat{x}, \Delta\hat{y}, \hat{\omega}, \hat{\eta})$, to the anchor box parameters, see Eq (13). Finally, a non-maximum suppression algorithm ranks the score $S$ of each potential region, removes regional overlaps (Fig 3D Bottom), and generates a set of neuronal regions $\{R_k\}$ (Fig 3E).

## Stage 3: 3D merging

In the third stage, we introduce X*NeuroAlignment* (Fig 4) to merge $\{R_k^z\}$ that belong to the same neuron. We introduce superscript $z \in \{1...Z\}$ to denote the frame index of a neuronal

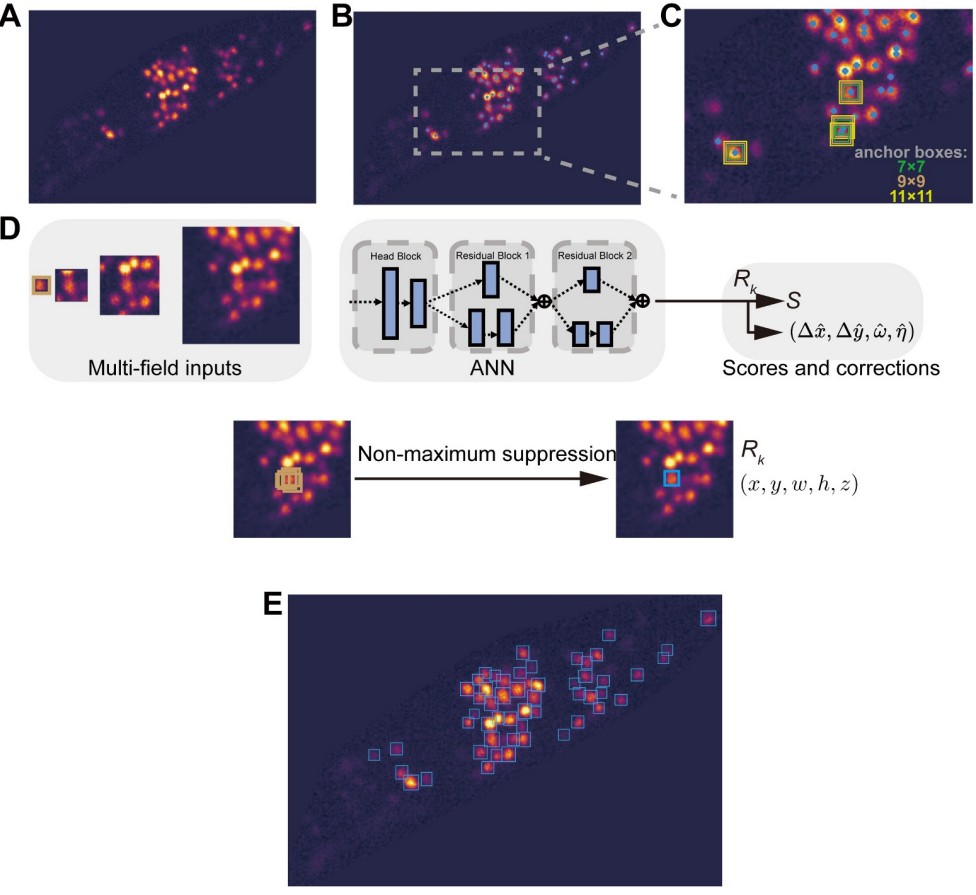

**Fig 3. Multi-Field Detection (MFD). (A)** A cropped head region. Every frame in a volume is cropped in the same way. **(B)** Find local pixel intensity maxima (blue points) in **A**. **(C)** An enlarged view of the rectangular region in **B**. Anchor boxes of different sizes are centered at 3 local maxima. The green, brown or yellow rectangle represents a $7 \times 7$, $9 \times 9$ or $11 \times 11$ anchor box, respectively. **(D)** Up: an artificial neural network (ANN) including residual blocks receives multi-field inputs once at a time and outputs a score $S$ and corrections $(\Delta\hat{x}, \Delta\hat{y}, \hat{\omega}, \hat{\eta})$ to the position and shape of an input anchor box. Bottom: ANN outputs several candidate bounding boxes surrounding an input local maxima, from which the optimal one is chosen by the non-maximum suppression algorithm. **(E)** A detection result (also see S3 Fig). Refined anchor boxes centered at local pixel intensity maxima are filtered by a non-maximum suppression algorithm. Blue rectangles are detected neuronal regions.

region. The continuous and sometimes irregular movements of a worm could lead to considerable shifts of $\{R_k^z\} \subseteq \mathcal{N}_i$ (Fig 4A), a major challenge for achieving accurate alignment. In general, stage 3 could be viewed as a maximum weight matching problem in a graph: each $\{R_k^z\}$ is represented by a vertex; vertices belonging to the same neuron are connected by an edge whose weight is determined by the overlap between the two neuronal regions, see Eq (2). This computationally intensive problem is significantly simplified by the following two observations.

- Exclusion Principle: each neuron appears at most once in a frame. All vertices representing $\{R_k^z\}$ with identical $z$ are disconnected.

- Continuum Principle: each neuron appears in only one frame or in consecutive frames. Other scenarios are forbidden. When a neuron appears in adjacent frames, an edge $E$ is defined to connect two neuronal regions:

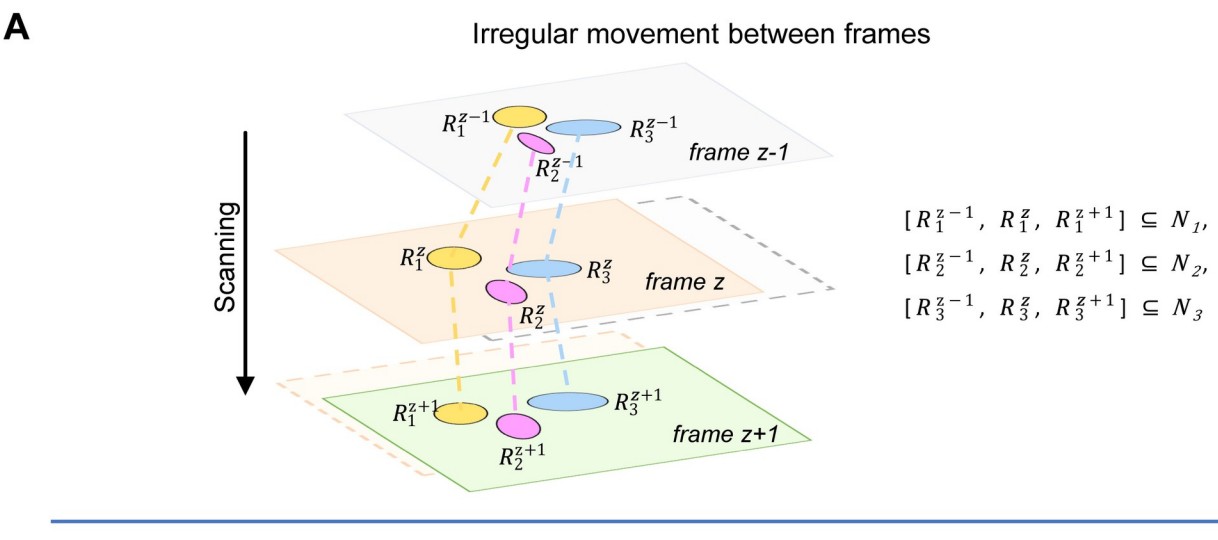

**Fig 4. 3D merging.** **(A)** An illustration of irregular cell movements across frames. The problem of regional alignment arises when the movements of neurons are significant during a single *Z*-scan of the imaging volume. Regions belonging to the same neuron, which are represented by ellipses with an identical color, could display considerable displacements between frames. The X*NeuroAlignment* algorithm for 3D merging: **(B)** The algorithm views neuronal regions in a volume as vertices in a linked list of bipartite graphs, where the IoU score between neuronal regions is the weight of an edge. **(C)** Every possible alignment between adjacent frames can be represented by a weight matrix, and the Hungarian algorithm is used to find an optimal row and column permutation that maximizes the trace (yellow color) of the matrix. **(D)** A 3D regional split algorithm calculates the mean pixel intensity $I$ of a neuronal region across all the frames in a neuron. If the intensity profile $I(z)$ has a local minimum at $z_0$, the algorithm will split the neuron object at $z_0$ (red vertical line).

$E \in \{(R_m^z, R_n^{z+1}) \mid z \in \{1 \dots Z-1\}, m \in \{1 \dots M_z\}, n \in \{1 \dots M_{z+1}\}\}$, where $M_z$ is the number of neuronal regions in the $z$th frame.

Therefore, the combinatorial optimization problem is reduced to finding a set of edges $\{E\}$ that maximize the total weights $\mathcal{W}$ in a chain of bipartite graphs:

$$\underset{\{E\}}{\operatorname{argmax}} \, \mathcal{W} = \sum_{z=1}^{Z-1} \underset{\{E^{z,z+1}\}}{\operatorname{argmax}} \, \mathcal{W}^{z,z+1}, \tag{1}$$

where $\mathcal{W}^{z,z+1}, \{E^{z,z+1}\}$ are total weights and edges in a bipartite graph connecting vertices from adjacent frames. Because each neuronal region belongs to only one neuron, Eq (1) is subject to the constraint that $\{E^{z,z+1}\}$ *should not* share a common vertex. In other words, the entire graph is a linked list (Fig 4B), not a tree.

We now define the weight of an edge $(R_m^z, R_n^{z+1})$ using the Intersection over Union (abbr. IoU),

$$w(R_m^z, R_n^{z+1}) = \frac{R_m^z \cap R_n^{z+1}}{R_m^z \cup R_n^{z+1}}, \tag{2}$$

which quantifies the regional overlap. Eqs (1) and (2) can be solved by the Hungarian algorithm (Fig 4C). Note that all $w(R_m^z, R_n^{z+1}) < \tau$ are set to 0, where $\tau = 0.05$ is a predefined IoU threshold.

The Hungarian algorithm will accidentally merge two neighboring neurons along the $Z$ axis. We design a partition method to resolve this problem. First, we compute the mean pixel intensity $I$ of every neuronal region $R_k^z$ in a neuron $\mathcal{N}_i$. Second, if the intensity profile $I(z)$ has a local minimum at $z_0$, we will split $\mathcal{N}_i$ at $z_0$th frame (Fig 4D).

Our partition algorithm relies on the observation that the displacement of the neuronal region $R_k$ is dominated by worm motion in the $X$-$Y$ plane. In our experiments, *C. elegans* was sandwiched between a flat agar pad and a coverslip, thereby restraining head movements in the $Z$ direction. Nevertheless, large up and down movements could lead to incorrect partition of a single neuron and unification of multiple neurons. One way to amend this problem is to impose an upper and lower bound on the longitudinal neuronal size. We notice that there is a strong correlation between the cell nucleus size in the $X$-$Y$ direction and in the $Z$ direction, suggesting that size information can be directly learned from data.

## Stage 4: Neuron recognition

In the final stage, we propose a few-shot learning method to recognize the digital identity of each neuron. First of all, we will discuss how to extract meaningful features that can help distinguish neurons. Second, we will introduce a deep recognition neural network for rapid inference of digital ID across imaging volumes of a single animal.

In the spirit of few-shot learning, we aim to carefully design neuronal features to facilitate machine learning when only a small number of human-annotated training examples is present. Here, we face two major challenges. First, a typical neuron in our imaging volume does not possess a distinct texture or shape, and the disparity in the appearances of the same neuron across volumes can be even bigger than those between different neurons (Fig 5A). Second, the location of a neuron is changing. What remains to be invariant and distinct across imaging volumes of an animal is a neuron's position relative to its surroundings (Fig 5B). We therefore propose two discriminable features based on density distribution surrounding a neuron and each neuron's $K$-nearest neighbors.

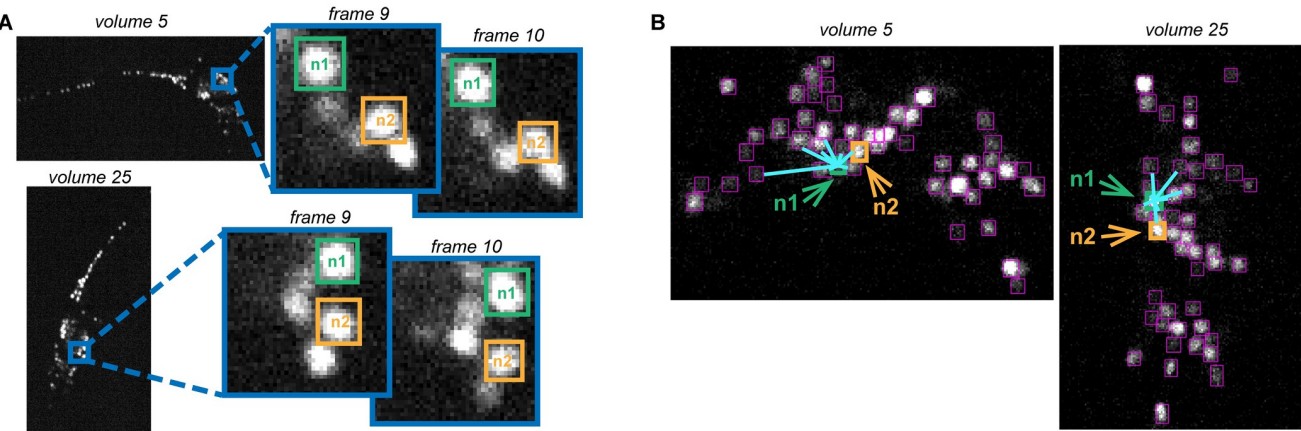

**Fig 5. A neuron's relative position is a more discriminable feature than its shape and texture.** (**A**) The shape or texture of a neuron in an imaging volume is not a distinct feature, which varies significantly across volumes. (**B**) The position of a neuron relative to its neighboring partners is a more robust and discriminable feature. For example, neuron *n1* and *n2* display similar relative positions in volumes 5 and 25.

**Neuronal density feature.** Let us provide a formal definition of the neuronal density function at a position $\mathbf{x} = [x, y, z]$ using the Dirac $\delta$ function

$$g_i(\mathbf{x}) = \sum_{\substack{\{R_j\} \\ R_j \notin \mathcal{N}_i}} \delta(\mathbf{x} - \mathbf{x}_{R_j}),$$

(3)

where $\mathbf{x}_{R_j}$ is the location of a neuronal region $R_j$ in our imaging volume, and the origin $[0, 0, 0]$ is the centroid of neuron $i$. Here and below, we ignore the size of a neuronal region by treating it as a point in 3 dimensions, and utilize the *C. elegans* coordinate system (see Stage 1: Pre-processing) to compute each neuron's relationship with its surrounding.

We can understand the physical meaning of $g(\mathbf{x})$ by integrating Eq (3) over a small volume $d^3\mathbf{x}$ at a separation $\mathbf{x}$ from the origin, which is identical to counting the number of neuronal regions in that small volume. $g_i(\mathbf{x})$ is also called pair correlation function in statistical physics. We introduce the subscript $i$ to emphasize that $g_i(\mathbf{x})$ is a different function when a different neuron $\mathcal{N}_i$ is centered at the origin of the *C. elegans* coordinate system.

In practice, directly discretizing Eq 3 generates a very sparse feature vector that is difficult for a neural network to learn (see below). We also need to take into account the observation that neuronal distribution is anisotropic: cells are extensively distributed in the *X-Y* plane but are confined in a narrow axial range. To extract meaningful statistics from data, we find it advantageous to design the feature vector in the cylindrical coordinate $[\rho, \phi, z]$, and to integrate $g_i(\rho, \phi, z)$ over a larger volume. We therefore count the number of neuronal regions *in between* two concentric cylinders, whose depth and radius are given by $(\mathfrak{z}, l)$, and $(\mathfrak{z} - \Delta\mathfrak{z}, l - \Delta l)$ respectively (Fig 6A). To capture density variations in the azimuth and depth directions, we further divide the volume of integration into 4 territories, anatomically corresponding to ventral-dorsal and left-right sides. Such a division is empirical, reflecting the information coding and sparsity trade-off in the density feature engineering.

Mathematically, let us define $\mathcal{C}(a, b; c, d; e, f)$ as the set of points in a volume with $a \leq \rho \leq b; c \leq \phi \leq d; e \leq z \leq f$. Let us also define neuronal regions as a point set $\mathcal{X} = \{\mathbf{x}_{R_j}, R_j \notin \mathcal{N}_i\}$.

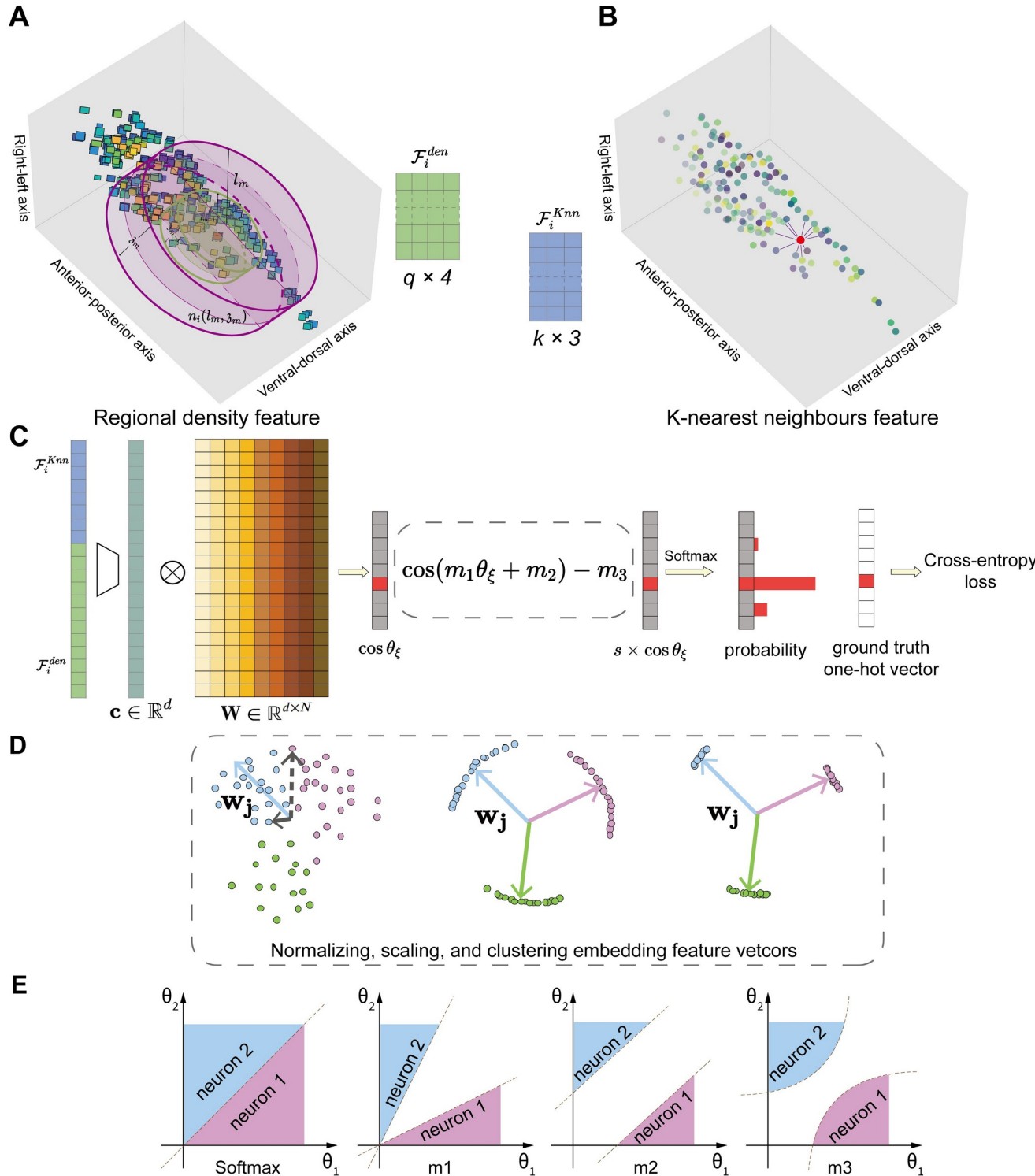

**Fig 6. Recognition of each neuron's digital identity. (A,B)** Neuronal feature engineering. In **A**, 3-dimensional atlas of all detected neuronal regions in a *C. elegans* head region. The red point assigns neuron $\mathcal{N}_i$ to the origin in the *C. elegans* coordinate system. The neuronal density feature is constructed by counting the number of neuronal regions $n$ between two color-coded concentric cylinders with expanding radii $l$ and heights $\mathfrak{z}$. The $n$ neuronal regions are subdivided into 4 territories (see Section Stage 4: Neuron recognition for details). The length of a density vector is $q \times 4$, where $q + 1$ is the number of concentric cylinders. In **B**, radiating lines connect $\mathcal{N}_i$ with its K-nearest neighbours (neuron objects), whose 3 coordinates are used as positional features. The feature vector is constructed by concatenating $\mathcal{F}_i^{den}$ in **A** and $\mathcal{F}_i^{Knn}$ in **B**. **(C)** A multi-class recognition neuronal network. First, the input feature vector is transformed into a high-dimensional embedding vector **c**. Second, the final linear layer computes the cosine similarity between **c** and each column vector

in the weight matrix **W**. Third, after incorporating margin penalty, scaling, and softmax operation, the model predicts the probability for each digital ID, from which a cross entropy loss between the model result and the ground truth is computed and back propagated during the training phase. **(D)** Geometric representation of embedding feature vectors. $\mathbf{w_j}$ is a column vector in the weight matrix **W**. In the case of softmax classification (left), examining the dot product between $\mathbf{w_j}$ (blue arrow) and an embedding feature vector that belongs to the same class (solid black arrow) or a different class (dashed arrow) suggests that the intra-class dot product could be smaller than inter-class dot product, resulting in classification error. Normalization and scaling drive the embedding feature vectors onto a hypersphere (middle); margin penalties help reduce intra-class distance while increasing inter-class distance (right). After training, every column vector $\mathbf{w_j}$ now points toward the centre of a neuron cluster. **(E)** Four examples illustrate decision boundary and margin in a binary classification task: a different parameter $m$ in the loss function specifies a different geometry of the decision boundary. Adapted from [26].

Counting the neuronal regions in a given volume, for example, can be defined as

$$
\mathcal{X} \cap \mathcal{C}(0, l; -\pi/2, \pi/2; 0, \mathfrak{z})
$$
$$
\equiv \int_0^{\mathfrak{z}} dz \int_0^l \rho d\rho \int_{-\pi/2}^{\pi/2} d\phi\, g_i(\rho, \phi, z) \tag{4}
$$

Counting the number of neuronal regions in the aforementioned 4 territories is thus given by

$$
n_i(l, \mathfrak{z}^+, V) = \mathcal{X} \cap \left( \mathcal{C}\left(0, l; -\frac{\pi}{2}, \frac{\pi}{2}; 0, \frac{\mathfrak{z}}{2}\right) \setminus \mathcal{C}\left(0, l - \Delta l; -\frac{\pi}{2}, \frac{\pi}{2}; 0, \frac{\mathfrak{z} - \Delta \mathfrak{z}}{2}\right) \right)
$$
$$
n_i(l, \mathfrak{z}^-, V) = \mathcal{X} \cap \left( \mathcal{C}\left(0, l; -\frac{\pi}{2}, \frac{\pi}{2}; -\frac{\mathfrak{z}}{2}, 0\right) \setminus \mathcal{C}\left(0, l - \Delta l; -\frac{\pi}{2}, \frac{\pi}{2}; -\frac{\mathfrak{z} - \Delta \mathfrak{z}}{2}, 0\right) \right)
$$
$$
n_i(l, \mathfrak{z}^+, D) = \mathcal{X} \cap \left( \mathcal{C}\left(0, l; \frac{\pi}{2}, \frac{3\pi}{2}; 0, \frac{\mathfrak{z}}{2}\right) \setminus \mathcal{C}\left(0, l - \Delta l; \frac{\pi}{2}, \frac{3\pi}{2}; 0, \frac{\mathfrak{z} - \Delta \mathfrak{z}}{2}\right) \right) \tag{5}
$$
$$
n_i(l, \mathfrak{z}^-, D) = \mathcal{X} \cap \left( \mathcal{C}\left(0, l; \frac{\pi}{2}, \frac{3\pi}{2}; -\frac{\mathfrak{z}}{2}, 0\right) \setminus \mathcal{C}\left(0, l - \Delta l; \frac{\pi}{2}, \frac{3\pi}{2}; -\frac{\mathfrak{z} - \Delta \mathfrak{z}}{2}, 0\right) \right)
$$

On the left hand side of Eq 5, $V/D$ indicates ventral/dorsal, and $\mathfrak{z}^+/\mathfrak{z}^-$ indicates left/right side of a worm respectively. The neuron density feature is a vector $\mathcal{F}_i^{den} = [n_i(l_1, \mathfrak{z}_1^+, V), \ldots, n_i(l_q, \mathfrak{z}_q^-, D)]$, where $l_q$ and $\mathfrak{z}_q$ are discretized quantities for $l$ and $\mathfrak{z}$.

Note that when applying the algorithm to datasets that do not possess neuronal region information, and include only the centroid coordinates of 3D neuron objects, we can straightforwardly modify Eq 3 into:

$$
g_i(\mathbf{x}) = \sum_{\substack{\{\mathcal{N}_i\} \\ j \neq i}} \delta(\mathbf{x} - \mathbf{x}_j), \tag{6}
$$

where we substitute $\mathbf{x}_{R_j}$ by the location of a 3D neuronal object $\mathbf{x}_j$.

**K-nearest neighbors feature.** The cylindrical coordinates $[\rho, \phi, z]$ of $K$-nearest neurons surrounding $\mathcal{N}_i$ are used to build the $K$-nearest neighbor feature vector (see Fig 6B): $\mathcal{F}_i^{Knn} = [\rho_1, \phi_1, z_1, \ldots, \rho_K, \phi_K, z_K]$. The $K$ neuron objects are arranged in an ascending order based on their distance from the origin.

**Recognition neural network, training, and loss function.** Next, a trained deep recognition neural network is used to predict each neuron's digital identity. The inputs to the recognition network is each neuron's feature vector, built by *concatenating* neuronal density feature and $K$-nearest neighbors feature from a given volume (Fig 6C). Below, we discuss model details.

In the final layer of our recognition neural network, an embedding feature vector **c** ($\mathbf{c} \in \mathbb{R}^d$) for neuron $\mathcal{N}_i$ is linearly transformed into an output **y** ($\mathbf{y} \in \mathbb{R}^N$), and the probability

of a given digital identity is computed using the softmax operation (Fig 6C):

$$\mathbf{y} = \mathbf{W}^T\mathbf{c}, \ \ \mathbf{W} \in \mathbb{R}^{d \times N} \tag{7}$$

$$p_j = \frac{e^{y_j}}{\sum_j e^{y_j}} \tag{8}$$

The corresponding cross-entropy loss $L_i$ for neuron $\mathcal{N}_i$ is given by:

$$L_i = -\log \frac{e^{|\mathbf{w}_\xi||\mathbf{c}|\cos\theta_\xi}}{e^{|\mathbf{w}_\xi||\mathbf{c}|\cos\theta_\xi} + \sum_{j \neq \xi} e^{|\mathbf{w}_j||\mathbf{c}|\cos\theta_j}} \tag{9}$$

where $\mathbf{w}_j \in \mathbb{R}^d$ is the $j$th column vector of the weight matrix $\mathbf{W}$; $\xi$ is the human-annotated label identity for $\mathcal{N}_i$; $\theta_j$ is the angle between $\mathbf{w}_j$ and $\mathbf{c}$; $\theta_\xi$ is the angle between $\mathbf{w}_\xi$ and $\mathbf{c}$.

The softmax loss function, however, is not optimized for intra-class similarity, namely the same neuron across volumes, and inter-class diversity, namely different neurons across volumes. An example in Fig 6D left shows how the spatial distribution of embedding feature vectors could lead to incorrect classifications. To improve the discriminability of the recognition model, we introduce several modifications in Eq (9) [26–32]. First, motivated by the hard examples in Fig 6D (left panel), we normalize $\mathbf{c}$ and $\mathbf{w}_j$ so that the loss function is exclusively computing the cosine similarity between a feature vector and a weight vector on a $d$-sphere. Second, to further increase the inter-class distances, we introduce a hyperparameter $s$ to scale the hypersphere radius (Fig 6D middle panel). Third, we introduce margin penalties $m$ as hyperparameters (Fig 6C) to maximize intra-class compactness and inter-class distance (Fig 6D right panel) in the $\theta$-space, the $N$-dimensional space spanned by the angles between an embedding feature vector and every column vector in the weight matrix $\mathbf{W}$.

Together, we consider the following loss function in our model

$$L_i' = -\log \frac{e^{s(\cos(m_1\theta_\xi + m_2) - m_3)}}{e^{s(\cos(m_1\theta_\xi + m_2) - m_3)} + \sum_{j \neq \xi} e^{s\cos\theta_j}} \tag{10}$$

The geometries of decision boundary in the $\theta$-space, specified by $m_1$, $m_2$, $m_3$ respectively, are different [26]. As an illustration, let us consider a binary classification task with two neuron classes (Fig 6E): $m_1$ penalty changes the slope of decision boundary; $m_2$ penalty translates the decision boundary in the normal direction; $m_3$ penalty makes the decision boundary nonlinear [26]. All three parameters would enforce a larger angular margin. By hyperparameter tuning, in practice we find that setting $m_1 = 1.05$, $m_3 = 0.05$ (S1 Table) gives the best performance in the validation data.

Fig 6D (right panel) provides a geometric view of the trained embedding feature vectors: $N$ discrete neuron clusters are distributed sufficiently far away from each other on a $d$-sphere with radius $s$. Because $\mathbf{w}_\xi$ is optimized to maximize its cosine similarity with $\mathbf{c}$, each column vector of $\mathbf{W}$ could well approximate a direction towards the centre of each cluster (also see S2 Fig).

**Inference.** During inference, the trained recognition neural network transforms concatenated feature vectors $\{[\mathcal{F}_i^{den}, \mathcal{F}_i^{Knn}]\}$ into embedding feature vectors $\{\mathbf{c}_i\}$, where the subscript $i$ indicates different neurons. Next, we consider the following two cases.

- Tracking within an animal: the test and the training volumes belong to the same animal. We compute the cosine distance matrix $D$ with $D_{ij} = 1 - \frac{\mathbf{c}_i \cdot \mathbf{w}_j}{|\mathbf{c}_i||\mathbf{w}_j|}$. The digital ID of each neuron can be allocated by finding the best match between all pairs, namely the optimal row and

column permutations that minimize the trace of *D*, a problem that can be solved by the Hungarian algorithm.

- Tracking across animals: the test and the training volumes belong to different animals. We first select an inference volume as a template, compute $\{c_j\}$ from the template, and let $\mathbf{W} = [c_1, \ldots, c_N]$. Next we also use the Hungarian method to allocate each neuron's digital ID in other inference volumes.

## Digital ID, cell-type ID, and human annotation

Throughout the paper, we have used and discussed two kinds of neuronal identities that worth formal definitions and clarification:

- *Digital ID*: a digital number assigned to a neuron. The number itself does not have any biological meaning. Because it is consistent across imaging volumes within an animal, digital ID allows the extraction of whole brain neural activity. digital IDs across animals do not have to correspond to each other.

- *Cell-type ID*: Uppercase letters indicating class based on anatomy and synaptic connectivity followed by L (left), R (right), D (dorsal), or V (ventral) [33]. Digital ID and cell-type ID do *not* have a fixed correspondence across animals.

Human annotation involves drawing bounding boxes to identify neuronal regions and assigning each neuron a digital ID across volumes within a single animal. A total of 350 volumes were labelled (C1, Table 1) in our main dataset. To identify neurons, we first label cells around the amphid and ventral nerve cord, for these two regions have relatively sparsely distributed neurons. Next, we utilize local structural traits to label neurons within the nerve ring. All labeled volumes are carefully examined and compared with a few pre-selected volumes. In order to test whether the knowledge learned by the machine can be used to track neurons in another animal, we also manually labelled volumes from two additional worms (C2 and C3, Table 1).

## Imaging setup

To capture whole brain neural activity in a freely behaving *C.elegans*, we combined a spinning disk confocal inverted microscope (Nikon Ti-U and Yokogawa CSU-W1, Japan) with a customized upright light path for worm tracking. Fluorescence signals, emitted from neurons at different depths in a head ganglion, were collected by a high NA objective (40X, NA = 0.95, Nikon Plan Apo), driven by a high-precision scanner (PI P721.CDQ). The measured lateral resolution is 0.30 $\mu$m/pixel; the scanning step along *Z*-axis is 1.50 $\mu$m. An imaging volume comprises 18 two-channel fluorescence images recorded at 100 fps by two sCMOS cameras

**Table 1. The organization of CeNDeR dataset.** To train and test CeNDeR, we used 3 human-annotated worms specified as C1, C2 and C3 in this paper. "/ vol" indicates an average number per volume.

| Dataset name | Worm name | Total volumes | Training volumes | Validation volumes | Test volumes | Number of labelled neurons /vol | Number of tracked neurons /vol | Strain |
|---|---|---|---|---|---|---|---|---|
| CeNDeR | C1 | 350 | n | 20 | 350 − (n + 20) | 161 | 161 | ZM9644 |
| | C2 | 30 | 0 | 0 | 30 | 142 | 142 | ZM9644 |
| | C3 | 30 | 0 | 0 | 30 | 137 | 137 | ZM9644 |

**Fig 7. The CeNDeR system. (A)** An RFP channel volume. **(B)** The CeNDeR pipeline takes the raw imaging data and outputs the location, shape and identity of every neuron in a volume. **(C)** A calibrated linear map allows the identification of each neuron's location in the green channel, from which the calcium fluorescence signals, $F_{\text{GCaMP}}$ of all recognized neurons can be extracted. **(D)** The whole-brain neural activity vector $\mathcal{R}$ can be inferred from the ratio of $F_{\text{GCaMP}}$ to $F_{\text{RFP}}$.

(Andor Zyla 4.2, England) simultaneously. Our imaging system thus has a volume rate $\approx 5$ Hz. Reducing the volume rate by increasing the inter-volume interval is not critical for CeN-DeR, since both our training and inference methods are sequence-independent. A customized infrared LED ring (850 nm) was mounted above a worm, and dark-field images of worm behaviors were captured by an upright light path and recorded at 25 fps by a USB-3.0 camera (Basler acA2000–165umNIR).

## Red fluorescence channel for detecting and tracking neurons

Our calcium imaging experiments were carried out on transgenic animals (ZM9644 *hpIs676 [Pregf-1::GCaMP6::3×NLS::mNeptune]*), where green fluorescent calcium indicator GCaMP and red fluorescent protein mNeptune are co-expressed in the cell nuclei by a pan-neuronal promoter. The activity-independent RFP channel was used to detect and identify neurons; the activity-dependent signal was extracted by linearly mapping the neuron position from the red to the green channel (Fig 7). We inferred neural activity from the ratiometric measure:

$$\mathcal{R} = \frac{F_{\text{GCaMP}}}{F_{\text{RFP}}}, \tag{11}$$

where the red fluorescence $F_{\text{RFP}}$ was used as a reference to reduce spurious signals arising from animal movements.

## Code availability

The CeNDeR pipeline as well as the human annotation and proofreading toolkit can be accessed at https://github.com/Wenlab/CeNDeR.

## Results

CeNDeR aims to rapidly detect all neuronal regions in a volumetric image, extract each neuron's 3D shape, and assign a consistent digital ID to each neuron across spacetime in a freely behaving *C. elegans*. The entire pipeline is divided into two parts: neuron detection task and neuron recognition task. To tackle neuron detection, we borrow deep learning methods from object detection, and use a trained neural network to predict 2D bounding boxes of all

neuronal regions within a stack of fluorescence images. Next, by taking into account the physical constraints on spatial distribution of neuronal regions, we apply the Hungarian algorithm to merge 2D regions into 3D neuron objects. Unlike 3DeeCellTracker and other related methods that use extensive image registration and segmentation algorithms, we strive to minimize pixel/voxel-wise computations in order to maximize detection speed.

To tackle neuron recognition, we hypothesize that each neuron's digital identity (see Digital ID, cell-type ID, and human annotation) is encoded by the density distribution of cell regions surrounding the neuron of interest and the relative positions between neurons. Using these spatial information as inputs, we train a deep neural network classifier from a small number of human-annotated examples to allocate each neuron's digital ID across volumes. Because the inference of digital IDs can be carried out independently for each volume, our recognition algorithm is a sequence-independent method.

The Results section is organized as follows. First, we report the performance of the neuron detection model. Because there is no publicly available raw imaging dataset with human-annotated neuronal regions, we restrict the evaluation to the three datasets (Table 1) collected by our imaging system. Second, we evaluate the performance of the neuron recognition model by considering two tasks. In the first task, a small fraction of volumes collected from a single freely behaving *C.elegans* are used for training, while the remaining volumes from the same animal are used for testing. In the second task, we test the performance of our trained recognition neural network on a different animal (either the same strain or different strains generated by a different lab). Finally, we compare the model training time, inference speed, and accuracy with other state-of-the-art recognition methods under the same benchmark.

## Neuron detection

Neuron detection consists of two stages, neuronal region detection and neuron merging. The algorithms are detailed in Stage 2: Neuronal region detection and Stage 3: 3D merging. We use precision, recall, and F1 score to quantify region detection and merging results, respectively. We chose the model whose performance on the validation volumes showed the best 2D F1 score during the training phase (see Table 2 for details). If the IoU (intersection over union) between a predicted region and a ground truth region is greater than a threshold 0.3, the predicted region is a True Positive (TP). Precision is the ratio of the number of TPs to the number of predicted regions, and recall is the ratio of TPs to the number of ground truth regions. F1 score is defined as the harmonic mean of precision and recall, namely $2/(\text{recall}^{-1} + \text{precision}^{-1})$.

**Region correction.** We applied a simple location transform to implement region correction, inspired by Girshick et al. [34]. During training, an ANN took an anchor box as an input and generated a score $S$, as well as corrections to the box position and size $(\Delta\hat{x}, \Delta\hat{y}, \hat{\omega}, \hat{\eta})$. The ground truth that has the maximum overlap with the box is the target. The ANN computed the binary cross-entropy loss for $S$ and L2 loss for position and size corrections. The targeted corrections are defined as:

$$\Delta x = (x_G - x_A)/w_A$$

$$\Delta y = (y_G - y_A)/h_A$$

$$\omega = w_G/w_A \qquad (12)$$

$$\eta = h_G/h_A$$

where $x_G, y_G, w_G, h_G$ represent upper left coordinates, width and height of a ground-truth (target) region; $x_A, y_A, w_A, h_A, z_A$ represent upper left coordinates, width and height, and frame

index of an anchor box. During inference, a box and its predicted corrections $(\Delta\hat{x}, \Delta\hat{y}, \hat{\omega}, \hat{\eta})$ were transformed into a neuronal region candidate $(x, y, w, h, z)$:

$$x = \Delta\hat{x} \cdot w_A + x_A$$

$$y = \Delta\hat{y} \cdot h_A + y_A$$

$$w = \hat{\omega} \cdot w_A \tag{13}$$

$$h = \hat{\eta} \cdot h_A$$

$$z = z_A$$

**ANN training.** Our detection neural network was trained by 30 randomly chosen volumes and 20 validation volumes from C1, and tested by 300 volumes from C1, 30 volumes from C2, and 30 volumes from C3. We used stochastic gradient descent with warm restarts and cosine annealing optimization strategy [35] to optimize network parameters and avoid gradient explosion. The detection model received inputs from 5-channel $41 \times 41$ concentric multi-field images (Fig 3D), including an anchor box and 4 cropped images with various sizes $(15 \times 15, 31 \times 31, 41 \times 41, 81 \times 81)$. Smaller or larger patch size would facilitate local or global feature learning respectively. A patch smaller than $41 \times 41$ was padded to $41 \times 41$; a patch bigger than $41 \times 41$ was resized to $41 \times 41$. The algorithm removed regions whose scores $S < 0.4$ and whose overlapping thresholds (derived from the non-maximum suppression algorithm) are less than 0.20. These values gave the best performance in the validation data. Pre-defined anchor box size was set to "$9 \times 9$" or {"$7 \times 7$", "$11 \times 11$"}. Table 2 presents performances from different anchor box sizes, in which "$9 \times 9$" represents the best trade-off between training time, inference speed and accuracy, consistent with the observation that 9 is the average regional size by human annotation (S1 Fig).

## Neuron recognition

The neuron recognition model aims to allocate consistent digital IDs for detected neuron objects across a 4D imaging dataset from a freely behaving *C. elegans*. Throughout this subsection, the word *recognition* is used interchangeably with the word *tracking*.

**Table 2. 2D neuron detection and 3D merging results.** During detection, two different sets of predefined anchor box sizes—"$9 \times 9$" or {"$7 \times 7$", "$11 \times 11$"}—are used. Three experimental 3D merging results are presented: merging of human annotated neuronal regions, where 2D detection performance is irrelevant and thereby represented by "-"; 3D merging of neuronal regions detected by an ANN. Box sizes of "7, 11" have slightly better 2D detection and 3D merging F1 score, but this combination exhibits lower inference speed and requires significantly longer training time than box size "9" does. "ms/vol" refers to milliseconds per volume.

| Task name | Anchor box size | CeNDeR C1 | | | CeNDeR C2/C3 | | | Training time | Inference time (ms/vol) |
|---|---|---|---|---|---|---|---|---|---|
| | | Precision (%) | Recall (%) | F1 score (%) | Precision (%) | Recall (%) | F1 score (%) | | |
| Region detection | 9 (2.7 $\mu$m) | 90.67 | 94.91 | 92.74 | 90.43 | 94.85 | 92.58 | 4.5 hours | 556 |
| | 7,11 (2.1 $\mu$m, 3.3 $\mu$m) | 90.58 | 95.78 | 93.11 | 89.93 | 96.39 | 93.05 | 8 hours | 1087 |
| Merging | - | 98.54 | 85.82 | 91.74 | 95.48 | 91.68 | 93.53 | - | 43 |
| | 9 (2.7 $\mu$m) | 84.31 | 82.87 | 83.58 | 84.58 | 86.89 | 85.72 | - | 50 |
| | 7,11 (2.1 $\mu$m, 3.3 $\mu$m) | 85.91 | 84.09 | 84.99 | 86.24 | 88.52 | 87.36 | - | 54 |

**Tracking neurons within an animal.**    To evaluate the performance of our algorithm, we first trained our recognition neural network on a small number of randomly chosen, human-annotated imaging volumes, 20 validation volumes, and then tested the model on the remaining volumes from *the same animal*.

Our algorithm is detailed in Stage 4: Neuron recognition. Briefly, the recognition neural network received a concatenated feature vector (Fig 6A and 6B) from each neuron, including a $k \times 3$ dimensional $K$-nearest neighbor feature and a $q \times 4$ dimensional neuronal density feature. Each input feature vector was embedded onto a $d$-dimensional sphere with radius $s$. $k,q,$ $d,s$ were optimized hyperparameters in the model (S1 Table). Furthermore, we introduced and optimized margin penalty hyperparameters (S1 Table) in the cross entropy loss (Eq 10) to maximize inter-class distances while minimizing intra-class distances.

First, we report the tracking performance on C1 (Table 1), a total 350 human-annotated sequential imaging volumes collected by our imaging system. Notably, trained embedding feature vectors formed $N = 163 + 1$ segregated neuron clusters with a large inter-cluster distance (Fig 8A). $N$ is a hyperparameter representing the number of neuron classes (see S1 Table), where $+ 1$ denotes an extra class. Dimensionality reduction (Fig 8B) reveals that the majority of validation and test samples were organized near the corresponding trained cluster centers. The top-1 recognition accuracy increases monotonically with the number of training volumes, reaching an average 88.25% with 30 randomly sampled training volumes (Fig 8C). Furthermore, the accuracy approaches 95.48% with 130 training volumes. Across the 200 remaining test volumes, the recognition accuracy fluctuates, including a few cases where the performance drops below 60% (Fig 8D). Examining the MIP images suggests that these instances correspond to rapid and deep head bends, when dramatic deformation caused some neurons to be much closer to other sets of neurons. This precluded an accurate inference since our recognition method relies on relative positional information to identify neurons.

Second, we report the tracking performance on the public dataset NeRVE ([11] and Table 3). With 130 training volumes, the top-1 recognition accuracy approaches an average 86.51% (Fig 8E), lower than the performance on C1. We suspect two differences in the datasets affecting the recognition accuracy: (1) fewer neurons were detected and annotated in NeRVE (Table 3); (2) NeRVE only includes the centroid locations of neuron objects, while C1 also contains neuronal region information (see Stage 2: Neuronal region detection and S2 Table).

**Tracking neurons in a different animal with a pretrained model.**    We next evaluate the performance of our algorithm when the recognition neural network, trained on 130 imaging volumes of C1, was tested on a dataset from *a different animal*.

First, we tested the tracking performance on C2 or C3, data collected from the same strain (Table 1) and by an identical imaging system. The top-1 inference accuracy drops to an average 66.70% across the 60 test volumes (Fig 9A and 9B). Second, we tested the performance on NeRVE, data collected from a different pan-neuronal imaging strain and by a different lab ([11] and Table 3), and found that the top-1 inference accuracy is 50.08% (Fig 9A and 9B). These results suggest that worm-to-worm and strain-to-strain variability has a strong impact on the performance of our recognition algorithm.

We asked whether the variability across different datasets is a major challenge for other neuron recognition methods. Indeed, when the fDNC model [12] was tested on our datasets, the inference accuracy drops to an average 41.51% (green violin plots in Fig 9B).

In Table 4, we summarize our within-animal-tracking and across-animal-tracking results and compare them with other recognition methods under the same benchmark, together with training time and inference speed. It is important to point out that 3DeeCellTracker cannot be directly applied to freely behaving *C.elegans*: an imaging volume must first be straightened and coarsely aligned to a common template.

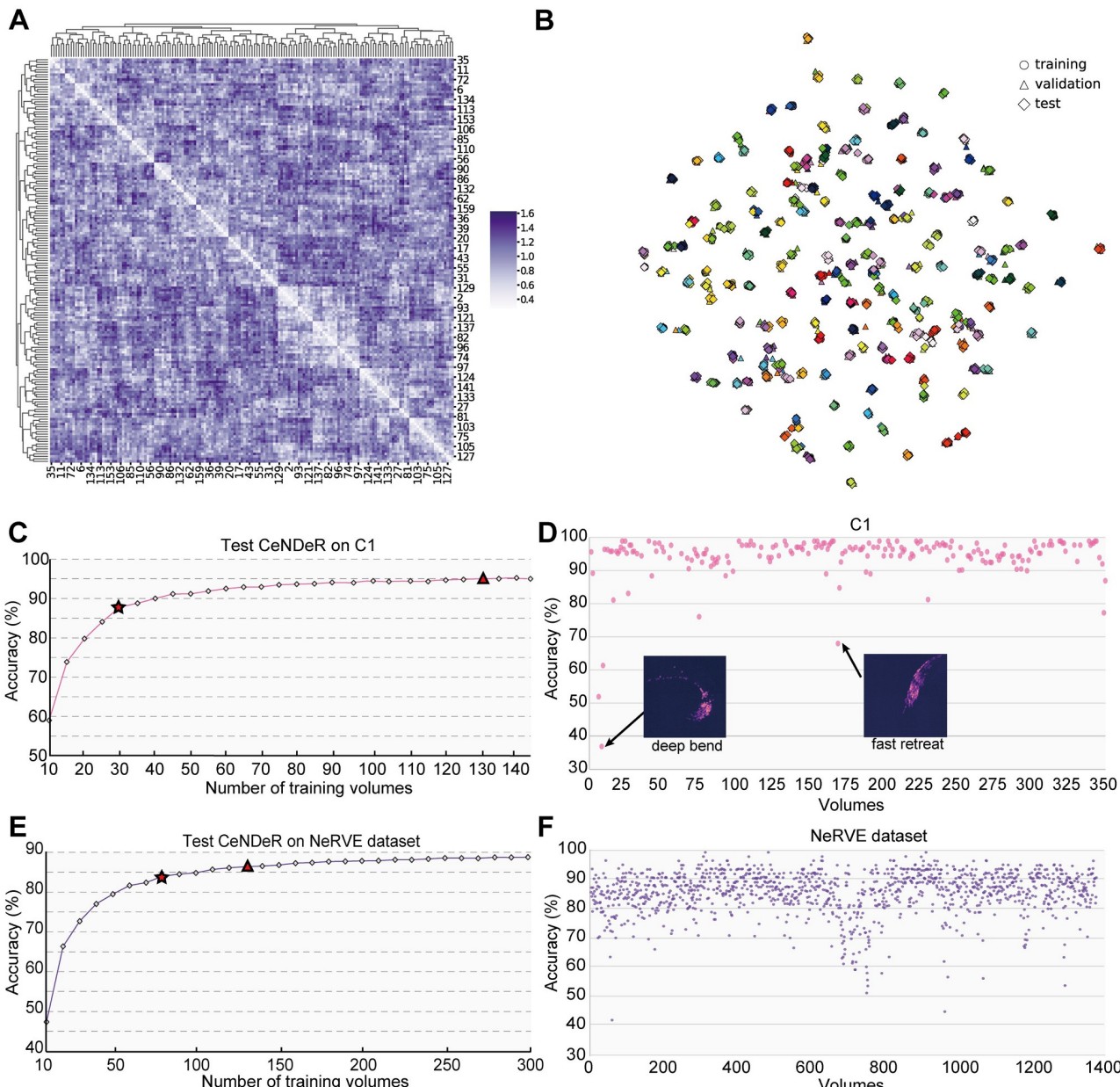

**Fig 8. Neuronal feature embedding and classification in the recognition model. (A)** Distance matrix between neuron clusters in the feature embedding space. The matrix represents the cosine distance $D = 1 - \frac{\mathbf{w_i} \cdot \mathbf{w_j}}{|\mathbf{w_i}||\mathbf{w_j}|}$ between trained weight vectors in the final layer of the recognition neural network (Fig 6C and 6D). $\mathbf{w_i}$ can well approximate the direction towards the corresponding neuron cluster center. (S2(B) and S2(C) Fig). The matrix indices, representing digital IDs, are rearranged by the hierarchical clustering method. $D_{min} = 0.28$, $D_{max} = 1.62$ when values along the diagonal are excluded, suggesting large inter-cluster distance. **(B)** t-SNE visualization of the feature embedding space. The low-dimensional representation reveals intra-class similarity (S2(A) Fig) and inter-class differences among neurons (related to **A**). For the sake of clarity, feature vectors from 4% of the training, validation and test volumes are drawn. **(C)** Train CeNDeR on C1 with different number of randomly chosen volumes and then test on the remaining volumes of C1. The top-1 accuracy reaches 88.25% with 30 training volumes (star), while the triangle denotes a 95.48% accuracy with 130 training volumes. **(D)** Overall tracking accuracy of C1 across 200 test volumes when the neural network was trained on 130 volumes. Inset shows MIP images of two cases with low recognition accuracy. **(E)** Train CeNDeR on NeRVE dataset with different number of randomly chosen volumes and then test on the remaining volumes of NeRVE. The top-1 accuracy reaches 84.10% (star) with 80 training volumes, the best performance among existing tracking methods (see Table 4). The triangle denotes a 86.51% accuracy with 130 training volumes. **(F)** Overall tracking accuracy of NeRVE dataset across 1222 test volumes when the network was trained with 130 volumes. In **D,F**, the horizontal axis represents the volume index over time.

**Table 3. Description of all Datasets.** All datasets used to train or test CeNDeR throughout this paper. The definition of Digital or Cell-type ID could be found in Digital ID, cell-type ID, and human annotation.

| Dataset name | State | ID type | Number of animals | Number of volumes | Number of labelled neurons /vol | Number of tracked neurons /vol | Strain | Reference |
|---|---|---|---|---|---|---|---|---|
| CeNDeR | Freely moving | Digital | 3 | 410 | 158 | 158 | ZM9644 | This paper |
| NeRVE* | Freely moving | Digital | 1 | 1372 | 69 | 118 | AML32 | [11, 12] |
| NeuroPAL Yu | Immobile | Cell-type | 11 | 11 | 58 | 134 | AML320(via OH15262) | [12] |
| NeuroPAL Chaudhary | Immobile | Cell-type | 9 | 9 | 64 | 119 | OH15495 | [21] |

* The actual dataset used is from [12], which has been rearranged into a more convenient form.

## Cell-type ID recognition

CeNDeR is *not* a method for identifying cell-type identities (see Digital ID, cell-type ID, and human annotation). Neurons with defined cell-type exhibit large positional variability across worms [17, 36], both in absolute positions along dorsal/ventral and anterior/posterior axis, and in pairwise distances between neurons (Fig 2 in [36]), a finding that is inconsistent with the central assumption behind our recognition algorithm. Indeed, when testing our recognition method on the NeuroPAL datasets (Table 3), which contain volumes from different animals with annotated cell-type IDs, the top-1 inference accuracy is merely 30% (Table 5). Therefore, additional information, such as color [17], is required, and must be combined with our current method to allocate cell-type IDs.

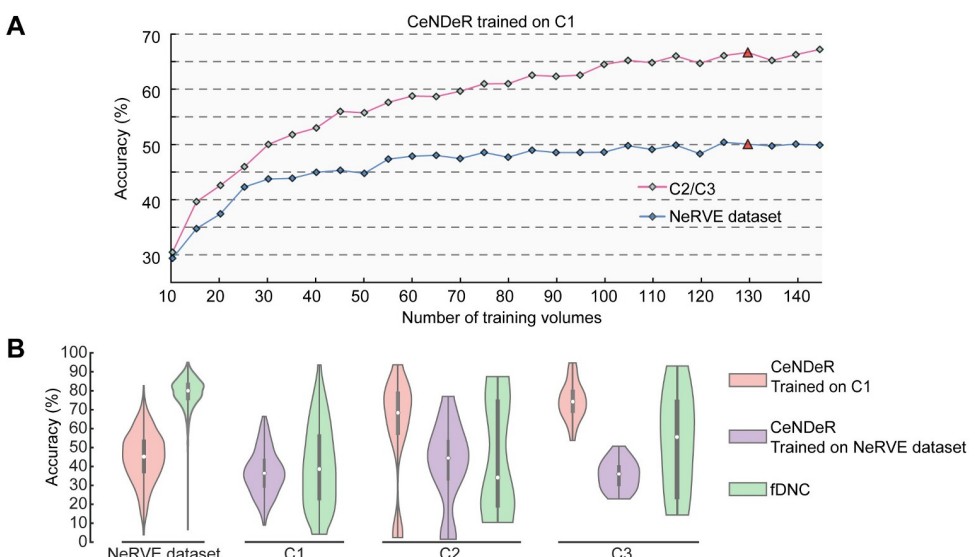

**Fig 9. Performance of tracking neurons in a different animal with a pretrained model. (A)** Tracking accuracy of C2/C3 and NeRVE dataset when CeNDeR was trained on C1 with different number of volumes. **(B)** Violin plot shows the distribution of tracking accuracy across animals. Pink colors: NeRVE dataset, C2 and C3 inferred by the CeNDeR method when CeNDeR was trained on 130 volumes from C1; purple colors: C1, C2 and C3 were inferred by the CeNDeR method when CeNDeR was trained on 130 volumes from NeRVE dataset; green colors: C1, C2, C3 and NeRVE dataset inferred by the fDNC method. Box plot inside each violin shows 25 and 75 percentiles of the distribution and the circle indicates the mean.

**Table 4. Tracking methods comparison.** We summarize the performances of four different methods in tracking neurons. The task is to assign neurons with consistent digital IDs across a 4-D imaging dataset. "s/vol" refers to seconds per volume.

| Task | Method | Condition | Test dataset | Accuracy (%) | Training dataset | Training volumes | Training time | Inference time(s/vol) | Model name | Reference |
|---|---|---|---|---|---|---|---|---|---|---|
| Tracking neurons within an animal | CeNDeR | Freely moving | CeNDeR C1 | 95.48 | CeNDeR C1 | 130 | 10 mins | 0.02 | M1♣ | This paper |
| | CeNDeR | Freely moving | CeNDeR C1 | 88.25 | CeNDeR C1 | 30 | 6 mins | 0.02 | M2♣ | This paper |
| | CeNDeR | Freely moving | NeRVE | 86.51 | NeRVE | 130 | 9 mins | 0.01 | M3♣ | This paper |
| | NeRVE(1)◇ | Freely moving | NeRVE | 73.1 | NeRVE | 1 | - | 10 | - | [11, 12] |
| | NeRVE(100)◇ | Freely moving | NeRVE | 82.9 | NeRVE | 100 | - | > 10 | - | [11, 12] |
| | 3DeeCellTracker (single mode)* | Immobile | Straightened NeRVE | 73 | NeRVE + synthetic | 1 + 500,000,000 | - | ≤ 73 | - | [13] |
| | 3DeeCellTracker (ensemble mode)* | Immobile | Straightened NeRVE | 99.8 | NeRVE + synthetic | 1 + 500,000,000 | - | ≤ 86 | - | [13] |
| Tracking neurons in a different animal with a pretrained model | CeNDeR | Freely moving | CeNDeR C2/C3 | 66.70 | CeNDeR C1 | 130 | 10 mins | 0.02 | M1♣ | This paper |
| | CeNDeR | Freely moving | CeNDeR C1/C2/C3 | 39.93 | NeRVE | 130 | 9 mins | 0.01 | M3♣ | This paper |
| | fDNC | Freely moving | CeNDeR C1/C2/C3 | 41.51 | Synthetic | 268000 | 12 hours | 0.01 | - | [12] |
| | CeNDeR | Freely moving | NeRVE | 50.08 | CeNDeR C1 | 130 | 10 mins | 0.02 | M1♣ | This paper |
| | fDNC | Freely moving | NeRVE | 79.28 | Synthetic | 268000 | 12 hours | 0.01 | - | [12] |

♣ Model hyperparameters and feature vector design can be found in S1 Table. If a dataset does not have neuronal region information, we constructed corresponding feature vectors from neuronal objects.

* 3DeeCellTracker takes 1 additional training volume from the experimental dataset. The reported accuracy covers only the first 300 volumes of the NeRVE dataset.

◇ NeRVE(1) uses one template for tracking; NeRVE(100) is the full NeRVE model that takes 100 templates from the same individual to make a single prediction ([12]).

**Table 5. Cell-type IDs recognition.** We summarize and compare the performances of CeNDeR with CRF and fDNC when tested on NeuroPAL strains. Selecting different template volumes would produce variable recognition results. Here, we present the mean and standard deviation of the recognition accuracy. We reevaluated fDNC results using their given model. "s/vol" refers to seconds per volume.

| Method | NeuroPAL Yu Accuracy (%) | NeuroPAL Chaudhary Accuracy (%) | Training dataset | Training vols | Training time | Inference time (s/vol) | Model name | Reference |
|---|---|---|---|---|---|---|---|---|
| CeNDeR | 27.79 ± 1.11 | 42.97 ± 2.35 | CeNDeR C1 | 130 | 10 mins | 0.02 | M1♣ | This paper |
| CeNDeR | 27.22 ± 1.35 | 39.82 ± 0.64 | NeRVE | 130 | 9 mins | 0.02 | M3♣ | This paper |
| CRF (open atlas) | - | 40 | Synthetic | - | - | - | - | [21] |
| CRF (data driven atlas) | - | 74 | Synthetic | - | - | - | - | [21] |
| fDNC | 57.84 ± 4.26 | 75.65 ± 3.15 | Synthetic | 268000 | 12 hours | 0.01 | - | [12] |

♣ Model hyperparameters and feature vector design can be found in S1 Table. If a dataset does not have neuronal region information, we constructed corresponding feature vectors from neuronal objects.

**Fig 10. Speed of the CeNDeR pipeline.** The processing time (ms/vol) of the entire pipeline and at each step. The detection step consists of finding local pixel intensity maxima and an ANN detection model. The recognition step contains feature engineering and a multi-class ANN model.

## Speed analysis

We present, in Fig 10, a detailed analysis of the processing time at each step in the CeNDeR pipeline. The two ANNs (detection and recognition neural networks) ran on a GeForce RTX 3090 GPU (24 GB, 29 TFLOPS in single precision) with CUDA 11.4, and other computations ran on an Intel Xeon Gold 5220R CPU (24 Cores, 2.2 GHz). Our analysis excludes the time for data loading and results saving. CeNDeR is a highly-encapsulated and parallelized system that is able to process acquired volumes simultaneously (e.g., 410 volumes in 5 minutes). Table 6 also compares the speed of CeNDeR pipeline with other methods.

## Discussion

CeNDeR is able to rapidly detect and recognize neurons (<1 sec/vol) from whole brain imaging data without compromising the accuracy (Tables 2 and 4). If speed is a concern, our stage 2 and stage 3 in the pipeline (see Stage 2: Neuronal region detection and Stage 3: 3D merging) provide an alternative to neuron detection methods that rely heavily on pixel- or voxel-wise segmentation of neuronal objects such as 3D-UNet [14] and watershed algorithm [13]. There are several other promising approaches for neuron detection. The one-stage deep-learning detection algorithm (e.g., YOLO [37]) may be a faster solution given that neurons in our dataset possess similar shapes and sizes; 3D detection or 3D instance segmentation (e.g., Mask R-CNN [38]) may be an end-to-end solution to detect or segment three-dimensional neurons directly.

Here a "9 × 9" anchor box (2.7 $\mu$m in size) was selected for neuron detection, reflecting a trade-off between training time, speed and accuracy. Since data collected by customized optical systems entail different pixel resolutions, a different lab may opt for a different box size to suit their best need. In addition, different axial resolutions can affect the performance of 3D merging and neuron recognition, for lower $Z$ resolution results in fewer neuronal regions. In the extreme case when only 3D neuron objects were used for feature engineering, as in the NeRVE dataset, we found that the accuracy of neuron recognition could be significantly different from that using neuronal region information (S2 Table).

**Table 6. Pipeline speed comparison.** We compare the speed of CeNDeR pipeline with other methods, which must tackle both detection and tracking tasks. "s/vol" refers to seconds per volume.

| Method | Inference time (s / vol) | Reference |
|---|---|---|
| 3DeeCellTracker (ensemble mode) | 86 | [13] |
| 3DeeCellTracker (single mode) | 73 | [13] |
| NeRVE | 48 | [11] |
| CeNDeR | 0.8 | This paper |

Our lightweight recognition model is suitable for tracking neurons over a long recording in freely behaving *C. elegans* and it complements state-of-the-art methods. Both fDNC and CeNDeR can rapidly track neurons at 10–20 ms/vol, but ours requires much less training time (Table 4). To use our method, we recommend the following steps to achieve the best performance.

1. Annotate carefully a small number ($\sim$30) of imaging volumes that cover different head postures among the recorded 4D imaging dataset.

2. Train the recognition neural network from scratch.

3. Perform inferences on the remaining imaging volumes.

While the overall performance of our recognition method is excellent on the within-animal-tracking task, there are worst-case scenarios when head bending and brain deformation are dramatic. Human annotation of 30 volumes may still be a non-trivial amount of user input. Furthermore, tracking neurons across animals remains an unsolved and challenging problem. Here we suggest several ways to improve our current algorithm with the aid of newly developed genetic, imaging, and computational methods.

- Whole brain calcium imaging has been augmented by sparse and combinatorial labeling of neurons that can be seen in orthogonal color channels [17, 36]. These landmark neurons possess invariant features and can be added to the existing feature vectors to improve the recognition accuracy. Color information could be very helpful in the scenario of large brain deformation, namely when head movement causes a large group of neurons to be squeezed into a very small region. In addition, combinatorial color labeling can be combined with our current method to provide cell-type ID information.

- In order to completely solve the neuron recognition problem under the umbrella of supervised learning, we, as a community, should make a great effort for sharing and standardizing datasets collected from different animals, different strains, and by different labs. Creating an equivalent *ImageNet* [39] database for *C. elegans* and other model organisms can leverage the power of deep-learning for reliably and accurately tracking neurons across animals.

- The idea of generating synthetic datasets [12, 13] that quintessentially capture the neuron movements and their variability across animals is appealing. Better algorithms can be developed in this direction. With sufficient supervised data, different ideas, such as the transformer [12, 40] and our feature embedding method, can be combined to further improve the recognition accuracy.

CeNDeR is a versatile method for detecting and recognizing neurons and their activity in a freely behaving animal. When dealing with data collected under different experimental conditions (e.g., different imaging systems and worm strains), our method aims to quickly learn the statistical representation of neurons and their relationships from a small number of human-annotated examples. We have also made an effort to minimize the number of parameters that need to be blindly tuned. Our supervised approach is potentially applicable to a spectrum of volumetric imaging data, including neural activity recordings in other model organisms, such as *Drosophia*, zebrafish, and mice.

## Supporting information

**S1 Table. Summary of tuned hyperparameters and feature vector design.** For a given training dataset, identical hyperparameters are used for feature engineering and network training. Neuronal regions (Region) or objects (Object) are used to build KNN and density feature

vector, respectively. *k* and *q* are parameters for the KNN and neuronal density feature, respectively. *d* is the dimensionality of the feature embedding space and *s* is the hypersphere radius. $m_1$, $m_2$ and $m_3$ are margin penalty coefficients. *N* is the number of neurons to be recognized.
(PDF)

**S2 Table. An exploration of feature vector design using neuronal region or neuronal object.** The recognition model was trained on C1 by using KNN and neuronal density as a concatenated input vector and tested on various benchmarks, from which the mean accuracy was calculated. 4 different combinations (neuronal regions or objects) were used to construct input feature vectors during the training stage, and we refer to M1 in S1 Table for the hyper-parameters. If the test benchmark does not contain neuronal region information, we constructed feature vectors using only neuronal objects.
(PDF)

**S1 Fig. Human-annotated region size distribution.** The median and average size is 9, so we design two types of anchor boxes: size 9 and size 7, 11. The second type would better cover regional size $\leq 6$ or $\geq 13$. See Table 2 for task performance.
(PDF)

**S2 Fig. Statistical analysis and visualization of the recognition model. (A)** Intra-class cosine distance distributions in every neuron cluster. *Y*-axis is cluster index; *X*-axis is the cosine distance between an embedding feature vector **c** and the corresponding weight vector $\mathbf{w}_\xi$ across the training set. A dashed line indicates a truncation to fit the figure panel into the page. **(B)** Distribution of cosine distance between a neuron cluster centroid and the corresponding weight vector $\mathbf{w}_\xi$ across the training set. Because the distance is very small (with a median $\approx$ 0.04), $\{\mathbf{w_j}\}$ could represent neuron cluster centers. **(C)** t-SNE visualization of neuron cluster centroids and weight vectors in the embedding space, related to **B**.
(PDF)

**S3 Fig. Detected neuronal regions with bounding boxes and digital IDs.** Detection (pink rectangle) and neuron tracking results (blue number inside a pink rectangle) of C2. Note that a raw imaging volume ($1024 \times 1024 \times 18$) has been automatically cropped into a smaller size ($273 \times 237 \times 18$) to embed a head region.
(PDF)

## Acknowledgments

We thank Wesley Hung, Min Wu and Mei Zhen for sharing an unpublished reagent ZM9644, Shangbang Gao for reading our manuscript. Request for the strain is directed to meizhen@lu-nenfeld.ca. We also thank Xiangyu Zhang and Siqiang Yang for proof-reading some of the imaging volumes.

## Author Contributions

**Conceptualization:** Yuxiang Wu, Quan Wen, Tianqi Xu.

**Data curation:** Yuxiang Wu, Shang Wu, Tianqi Xu.

**Formal analysis:** Yuxiang Wu, Quan Wen, Tianqi Xu.

**Funding acquisition:** Quanshi Zhang, Quan Wen.

**Investigation:** Yuxiang Wu, Shang Wu, Xin Wang, Chengtian Lang, Quanshi Zhang, Quan Wen, Tianqi Xu.

**Methodology:** Yuxiang Wu, Xin Wang, Quanshi Zhang, Quan Wen, Tianqi Xu.

**Project administration:** Quan Wen, Tianqi Xu.

**Resources:** Quan Wen, Tianqi Xu.

**Software:** Yuxiang Wu, Tianqi Xu.

**Supervision:** Quan Wen, Tianqi Xu.

**Validation:** Yuxiang Wu, Shang Wu, Quan Wen, Tianqi Xu.

**Visualization:** Yuxiang Wu, Quan Wen, Tianqi Xu.

**Writing – original draft:** Yuxiang Wu, Quan Wen, Tianqi Xu.

**Writing – review & editing:** Yuxiang Wu, Quan Wen, Tianqi Xu.

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
