## [Decision Letter · Decision Letter 0]

6 Apr 2022

Dear Dr Wen,

Thank you very much for submitting your manuscript "Rapid detection and recognition of whole brain activity in a freely behaving Caenorhabditis elegans" for consideration at PLOS Computational Biology.

As with all papers reviewed by the journal, your manuscript was reviewed by members of the editorial board and by several independent reviewers. In light of the reviews (below this email), we would like to invite the resubmission of a significantly-revised version that takes into account the reviewers' comments. Please pay particular attention to reviewer comments concerning evaluation and benchmarking of the performance of your approach.

We cannot make any decision about publication until we have seen the revised manuscript and your response to the reviewers' comments. Your revised manuscript is also likely to be sent to reviewers for further evaluation.

Sincerely,

Blake A Richards

Associate Editor

PLOS Computational Biology

Daniele Marinazzo

Deputy Editor

PLOS Computational Biology

Please pay particular attention to reviewer comments concerning evaluation and bench-marking of performance of your approach.

Reviewer's Responses to Questions

**Comments to the Authors:**

Reviewer #1: Wu and colleagues present a novel and innovative method called CenDer that uses a set of algorithms to detect neurons in freely behaving C. elegans and track them across video frames in 3 dimensional volumetric images. These particular challenges were recently addressed by two innovative methods that were published in 2021 eLife papers. One method called 3DeeCellTracker was published by Wen C and colleagues in the Kimura, Hillman and Nemoto labs. Another method called fDNC was published by Yu X and colleagues in the Leifer and Linderman labs, who focused only on the challenge of tracking neurons in moving animals. In this new manuscript, Wu and colleagues use a combination of artificial neural networks and embeddings to achieve high accuracy on the tasks, but it seems they ultimately underperform 3DeeCellTracker which remains the state of the art. All that being said, the innovations brought forth by CenDer in this manuscript provide a new perspective on the existing problem and novel ideas that might be combined with the previous works to further improve accuracy for the challenge of detecting and tracking neurons. I believe that PLoS Computational Biology is an ideal venue for this manuscript and that readers will appreciate the ideas and innovations that are introduced by the authors.

The manuscript is nearly done and I suggest mostly minor revisions but I have three concerns that should be addressed. My first concern is that some sections lack clarity. Given how complex some of the ideas are the confusing parts of this manuscript should be rewritten for clarity. This will help researchers to reuse the algorithmic advances in future work to attack similar problems. My second concern is that the CenDer program is provided in several pieces that were written in different programming languages and requires a lot of work to use at the moment. Since this manuscript introduces and end to end solution for detecting and tracking neurons, the software should be made available in such a way that users can easily run it on prepackaged example datasets and also use it for their own datasets. One way to do this is to build a jupyter notebook for it, and an alternative way is to compile it and add a simple user interface. Lastly the performance comparisons to 3DeeCellTracker and fDNC use their reported values instead of comparing them on the same benchmark. For this reason it’s impossible to compare all three methods to determine which is best. A true head to head comparison on the same benchmark should be added so that readers can decide what solution to choose among the now three main choices.

1. Abstract: C. elegans is not a “mollusc”, the phylum is Nematoda. Same for Line 8.

2. Line 32-40: references should be added and addressed for “Simultaneous recognition and segmentation of cells: application in C. elegans” by Qu L, et al, Bioinformatics 2011; “A probabilistic atlas for cell identification” Bubnis G, et al, arXiv 2019; and “Probabilistic Joint Segmentation and Labeling of C. elegans Neurons” by Nejatbakhsh A, et al MICCAI 2020.

3. Line 42: “interference” should be “supervision and correction”.

4. Line 53: in “CenDer”, why is “Detection” capitalized but “neuron” and “recognition” are lowercase? I suggest capitalizing the remaining two words and rename it CeNDeR.

5. Introduction: 3DeeCellTracker and fDNC are the current state of the art programs for neuron recognition and tracking but they're not explicitly mentioned in the introduction. Readers deserve a better explanation of the state of the art programs available now, more information on the algorithms used by these programs, and a more explicit explanation of how CenDer improves upon or adds to the current state of the art work.

6. Line 71 and 123: what is “F”, Frame? Figure 1C uses (x,y,w,h,z). The authors should unify their notation and use “Z” in place of “F”. Or D and d, for depth, in case they think Z will be confused for integers and f for functions.

7. Fig 6D: the dashed colored lines barely visible and should be made thicker.

8. Line 84: is N a free parameter set by the user or is it fixed for the head? It looks like N was chosen from Table S5 but the authors should clarify this.

9. Algorithm 1, Require: how is the head contour determined? I see it’s the largest component after dilation but this should be made much more explicit to the reader.

10. Algorithm 1, procedure 5: how is the anterior-posterior directionality of the head determined? The algorithm generates a vector but how is the anterior direction of the worm determined?

11. Algorithm 1, procedure 6: how does the dorsal-ventral axis assignment work if the worm is twisted? It’s also unclear how the perpendicular angle is set to ensure it lies in the dorsal-ventral as opposed to any other in this axis, for example aligning instead with the left-right axis or some other angle between dorsal-ventral and left-right axes. The central moment is only sufficient to capture the anterior-posterior vector but the worm is a 3-dimensional deformable and twistable tube.

12. Line 91: why is the dilation filter larger than erosion? Please be explicit as to why empirical choices were made.

13. Line 110: please add a link to the results section for the ANN and specify that the details of its training/test set and validation are provided in that section.

14. Fig 3D: this panel is confusing, repetitive, and doesn’t really illustrate this portion of the algorithm. Please show just one iteration of the ANN architecture, then illustrate how multiple ANN outputs are unified into single a neuron detection. Since PLoS Computational Biology targets a broad computational audience, it would be best to imagine a naïve reader and then help them understand this process as best possible.

15. Line 150: the underlying assumption in the partition algorithm is that during successive frames in an image volume, neural vertices can be displaced due to worm motion within the XY plane, but not up and down in Z which could then lead to mis-partitioning and unification of multiple neurons into one very long neuron, or mis-partitioning one neuron into several smaller pieces. The authors should address this explicitly. One way to deal with this issue would be to impose an upper and lower constraint on neuron size that is learned from the dataset.

16. Line 152: what algorithm was used to determine if pixel intensity displays multi-modality?

17. Lines 163-180: this section is confusing and should be rewritten for clarity. What’s missing is both an overview, and also the motivation behind the various computations that take place. I assume the Dirac delta is used to approximate each neuron as a point but this should be made explicit. Why do the authors switch to cylindrical coordinates, is this for the purpose of computing the cosine similarity in the ANN? Why is the data sectioned into 4 quadrants? What happens when the head bends in opposite directions, across successive image volumes, such that some neurons are now closer to other sets of neurons and the KNN computation reallocates which neurons end up in which clusters?

18. Lines 195-203: this section is also confusing. The authors state that several modifications are performed to improve the discriminability of the model. These modifications should be covered in greater depth. I assume they’re all summarized as making space between the features in their embedding to make them more robust to a variety of noise in the feature vectors? It’s unclear why these choices were made and how they remedy the issues presented by real world videos. The authors need to improve their explanation here.

19. Line 188: what data are included in feature vector c, is it just the KNN values? This should be made explicit.

20. Stage 4, Neural Recognition section: this is a very difficult read and it’s one of the more novel stages of the algorithm. As is it doesn’t provide a good overview of what’s to come, the purpose of each part as it pertains to the whole isn’t clear, and the transitions between each part aren’t clear either. The authors would greatly benefit from having a naïve reader go through this section to help them in rewriting it to improve its clarity. This is also true of some of the previous sections, but to a lesser extent.

21. Line 218: both Toyoshima Y, et al BMC Biology 2020 and Yemini E, et al Cell 2021 show that neurons in C. elegans are positionally variable and thus require additional techniques, such as color, to identify cell types such as AWA. Assigning neuron cell-type ID’s across worms, using just the cell positions, leads to a considerable amount of mislabeling errors. The authors should address this and limit their claims to tracking neurons within single animals across image volumes. Neuron IDs in this paper appear to be limited to enumerations of N_i where i=1, …, total neurons. The authors should be explicit in explaining that their method can be used with other techniques for neural ID but cannot perform the actual cell type ID itself.

22. Line 231: as is the standard for such publications, the authors should provide the x, y, and z resolution for their microscope in microns. This becomes especially important later on when discussing neuronal regions in pixel “box” sizes, since these are expected to be sensitive to microscope resolution. In addition to this, the algorithm is likely limited to a minimum nuber of volumes per second below which it can no longer track neurons across frames. The authors should state the minimum volume rate needed for their algorithm to work.

23. Table 1: 80 total training volumes seems a bit low for training an ANN, I assume there are many neurons in each volume. The table should state how many neurons are present in each set of volumes to give the reader a better sense of whether sufficient observations were used to train the ANN.

24. Table 2: anchor box sizes are measured in pixels but the relevant measure is microns since different microscopes and objectives will result in images with different resolutions. The authors should provide box sizes in microns, in addition to their pixel measurements. The authors should also discuss the minimum axial and temporal resolutions required for their algorithm to work and whether they recommend downsampling high resolution images to improve their algorithmic speed and accuracy. This is especially relevant given their highly specific choices for anchor box and cropped image sizes, and choice of ANN training dataset, which are likely to depend strongly on volumetric resolution.

25. Lines 288-289: two thresholds, 0.4 and 0.2 are mentioned but seem arbitrary. The authors should clarify how these were chosen.

26. Line 307: it’s not clear why the link to Table S3 is given. The authors should clarify that this supplemental table shows why the values of d=56 and s=16 were chosen.

27. Table 3: as mentioned earlier, cross animal neuron ID is not verifiable without a method to perform cell type ID. The authors should remove this claim or use datasets that permit cell type ID to verify their claims. The Toyoshima Y et al 2020 and Yemini E et al 2021 papers make their real animal datatsets public. Wen C et al, and Yu X et al, both in eLife 2021 use both synthetic and real animal datasets to quantify the performance of their 3DeeCellTracker and fDNC algorithms, and these datasets are also publicly accessible. The authors can use any of these datasets to validate claims of cross animal ID if they wish to include this claim. That said I don’t see the claim of cross animal ID as necessary for this paper to be novel or of high impact, it is already very strong without this extra work.

28. Table 3, Accuracy: the comparison here uses reported accuracies from different papers using different benchmarks. It’s not an appropriate comparison. All algorithms should be tested on a common benchmark so that they can be compared head to head to determine which is best on each task. The authors also fail to report the published top-3 accuracy of fDNC which is 91.3% but this will no longer matter when all algorithms are tested on a common benchmark dataset. The authors should compare CenDer to 3DeeCellTracker, fDNC, and CRF by Chaudhary S et al, eLife 2021. A benchmark dataset was already created in the 3DeeCellTracker paper. The 3DeeCellTracker benchmark reused datasets from Toyoshima Y et al 2016 and Nguyen JP et al 2017 papers, and then added its own dataset to these. So the standard exists to use the existing benchmark datasets. The authors can add their own benchmark dataset to the existing ones if they feel that their dataset adds information that is relevant to the comparison.

29. I can’t verify that the software and algorithms work because they require a combination of python, matlab, and even Nvidia GPUs. I believe many readers will be unable to use this paper’s innovation for the very same reasons that I can’t. PLoS Computational Biology policy states “For studies that describe software for end-user applications, deposition either in a repository that enables remote code execution or instructions for installing and using the software are a prerequisite.” As is the code doesn’t conform to this standard. Since the code is largely in python and the matlab parts can be compiled into an executable, the authors should create a jupyter notebook that executes their code, provide a tutorial and examples to use it in the notebook, and allow users to upload their own volumetric images to the notebook so they can test these with the software.

Reviewer #2: Wu et al. describe CenDer, an interesting and speedy addition to the suite of algorithms for tracking of C. elegans neurons in whole-brain or whole-animal calcium imaging. This is a challenging problem for a variety of reasons, and is a very important one to solve well in order to allow biological insight to be derived from such recordings. In brief--we can watch worms think, but if we cannot keep track of which neuron is which even within an animal, much less across animals, we cannot understand what we are seeing.

The manuscript is clearly written, has figures that helpfully illustrate less-intuitive parts of the algorithm (e.g. concentric images), and sets out the current state of the art amply while describing the advances made by this paper. The main advance is that the method achieves quite-good performance with very little computation time--other approaches are either fast, but so poor as to be unusable (though to be fair it is not clear that the accuracy of this method is good enough either), or of good quality but slow enough to pose a major computational bottleneck. This paper's method computes neuronal identities only 4x slower than acquisition speed, making it very unlikely to be a bottleneck in any real setting. The best competitor, 3DeeCellTracker is 100x slower than CenDer and 400x slower than acquisition, meaning that an experiment that takes just three minutes will require a whole day to compute. Although this can be ameliorated somewhat by running on a large cluster or in the cloud, this emphasizes that speed is in practice a significant concern.

The authors do not, in this paper, present a final solution to the problem: though their method is fast and accurate, for some applications it is unlikely to be accurate enough. This manuscript should therefore be viewed both as a potentially useful tool and as a report on a promising direction that the authors or other groups could continue to pursue until accuracy is adequate. (The authors describe some potential improvements in the discussion.)

The biggest flaw in the results is that there does not appear to be any validation on a recording from an animal that has not been used as part of the training set. Instead, every animal is used to contribute to both training, test, and validation data sets by splitting the recording in time. While this is valuable, it is far from adequate. Even if it is too labor-intensive to annotate enough individual animals to get ground-truth for a reasonable sample size, showing at least one new animal is very important. Or, alternatively, the authors should at least retrain their network with two out of the three animals, and use the third annotated animal as the test set. The neuronal clustering appears promising--but this may be partly illusory due to the selection criterion for which neurons were even allowed to exist (i.e. only those appearing in all three--a retraining must be done WITHOUT excluding missing neurons from the held-out animal!). If I am mistaken and this never-trained-on-animal data does exist already, it should be highlighted much more, as this is how the tool would be actually used: people would generate new data sets, not annotated by humans, and would run the software to get their results. For the biologist, nothing else really matters. As a computational technique, even if it turns out that this doesn't work, the paper still has merit for its clever and speedy preprocessing and feature construction which provides an appropriate level of detail as input to the ML part of the analysis pipeline.

In addition to this one main concern--hopefully an easily addressed one--there are a variety of minor comments below.

Abstract: C. elegans are NOT molluscs! Their phylum is Nematoda!

Author summary: it would be more grammatically correct to say, "the fluorescent change in and thereby..." (add in). Alternatively, the sentence could be restructured.

Introduction: C. elegans are still NOT molluscs!

Methods, Stage 1: The tense is wrong in "We design an Automatic Pre-Processing (APP) algorithm". This happened in the past: "designed". The running of it can be stated in present tense (e.g. "to extract a head region and build a C. elegans coordinate system"..."use it to" is redundant and actually a bit confusing).

Methods, Stage 2: Here especially, but also elsewhere, the authors do not do a good job distinguishing between the *training* of an ANN and the *running* of the trained ANN to produce a result. Here, where the authors are describing the processing pipeline, "training" is not what is happening--the ANN is already trained. Thus, "Next, an Artificial Neural Network (ANN) is trained..." is not the right thing to say. The authors should clearly separate the two tasks:

(1) We trained an ANN on such-and-so data to find the center and size of candidate regions.

(2) We use this trained ANN as part of the processing pipeline.

Methods, Stage 3: It is very difficult to refer to the NeuroAlignment step due to the unexpected fancy calligraphic X as the first character (it's not a LaTeX standard that I'm familiar with, but looks similar to fonts I do know). It would be nice if the authors could just call it XNeuroAlignment or something like that, so that it's completely clear what the first letter is.

Methods, Stage 4: The K-nearest neighbors feature sectoin says "Here we ignore the size of a neuron object by treating it as a point in three dimensions." But this is what we've already been doing all throughout the Neuronal density feature section! That's the whole point of the delta function. I do also wonder about the formalism--while the integrals are technically correct, the entire problem is discretized and really is a point-set problem. Thus, I wonder if it might not be simpler to cast the problem as an intersection between the point coordinate set X = {x_R_j} and a set of concentric partial shells about a neuron i, which you can form by defining the partial volume V(i, h, l) and then computing X ^ (V(i, h, l) - V(i, h - dh, l - dl)). Anyway, treating neurons as points is not new by the time we hit K-nearest neighbors.

Methods, Imaging Setup: "inverted microscopy" should be "inverted microscope".

Methods, Red fluorescence channel...: Using the Fraktur font R, typically used to denote the real part of a complex-valued variable, is a peculiar choice. I understand that R is already in use for region, but this is liable to be confusing to any mathematician. Perhaps use Q, a calligraphic R instead of the Fraktur (\\Re) one, or something else. Every time I see it, I think that, for instance, neurons are complex-valued.

Figure 7: Same thing about Fraktur R used as ratio.

Results, ANN training: It would be nice if the authors say something about how long training takes, not just running the trained model. It seems like it ought not take very long, given the limited size of the data set, but I couldn't find any numbers. Also, in the pre-defined anchor box size line, 9x9, 7x7, and 11x11 sizes in quotes have close-quotes on both sides. Either they should be plain quotes (neither open nor closed), or they should be a proper open-closed pair. (Yes, I know LaTeX will tend to make this mistake at times.)

Figure 9: The colors of the different steps are too similar. Please make them more dramatically different so it's easy to tell at a glance what step is what. For instance, they could all be blue of different darknesses (substantially different!), with the preprocessing in green, if that is to stand out.

Reviewer #3: Wu and colleagues present a complete analysis pipeline for extracting calcium signals from large neural population recordings of moving C. elegans. Their pipeline segments neurons, tracks neurons through time, extracts calcium signals, and identifies the correspondence of neurons across animals. There has been a recent flurry of work in the literature presenting computational methods to address different aspects of this problem, which speaks to the interest and need in the community for developing a robust, high performing pipeline that could be of general use to the field.

My primary concern with this work is that the ground-truth evaluation of performance, as currently presented, is not yet satisfactory. If the reported performance holds up under more rigorous assessment, then I think this work would be a very valuable contribution to the field.

Most critical concerns:

1) Tracking performance (within-animal) should be more meaningfully assessed. Currently, performance is evaluated on an extremely small test set of 5 or 10 consecutive volumes representing less than 1 or 2 seconds of animal movement at the reported volume rate. This is woefully insufficient to evaluate the algorithm’s performance on a representative sampling of animal poses or movements. Evaluation on longer recordings will also help differentiate the performance of this algorithm from sequence-dependent methods (Lagache et al., 2021) that perform well on short times, but struggle with longer recordings.

2) The quality of ground truth used to evaluate across-animal neural ID performance is unsatisfactory. Other recent works (refs 19-21) have used human annotations of NeuroPAL strains to establish ground truth neural ID across animals. There a human relies on genetically encoded color labels in addition to position and morphology. In this work, it seems that ground truth is determined by a human using only position and morphology. I am skeptical of a human’s ability to accurately identify neurons across animals via position and morphology alone. More evidence is needed to show that the ground truth across-animal neural correspondence used here for evaluation is accurate.

Other major substantive feedback:

- The work would be strengthened by evaluating performance on another dataset from a different group. It is becoming clear that imaging hardware, conditions and labeling approaches vary across groups. If this work is to have the maximal impact on the field, it should show how well it can generalize to another C. elegans dataset. There are now at least a couple publicly accessible datasets in the literature. This also helps facilitate direct comparisons to other methods, because they can be evaluated on the same datasets.

- Please clarify when training must be performed, and all the instances where human input is required. For example, does the algorithm need to be trained anew for each worm? For each imaging setup? For each age of animal used or each strain? Make this information prominent in the text.

- Clarify how performance of this method is compared to competing methods in Table 3. Did the authors re-run those algorithsm on a common dataset? If not, where did the values listed come from? If they came from the literature, then please double check. Some of the numbers listed do not match what I found in the literature.

- Neuron detection superficially appears to have some similarities to 3DeeCellTracker (Wen et al., 2020). Please expand on similarities and differences.

Textual, visualization and some minor concerns:

- The detailed description of the pipeline can be hard to follow and reads more like a recipe. It would benefit from more high-level description to help guide the reader along and to better motivate each algorithm or implementation detail.

- Figure 9 is hard to decipher because all the colors are similar and I cannot tell which bar is which. Please remedy.

- Terminology is sometimes confusing. Please clarify when cell ID means ID across animals (e.g. the name in the connectome) vs the local identity within a worm.

- C. elegans come from phylum Nematoda not Mollusca . Remove all references to molluscan and mollusc.

- Some language is odd. “Gadgets” in “Despite the presence of modern gadgets” is wrong.

- 295: "A model was saved if it's performance has best" Please clarify. Does this refer to initializations of the parameters? Which are the different models?

**Have the authors made all data and (if applicable) computational code underlying the findings in their manuscript fully available?**

Reviewer #1: Yes

Reviewer #2: Yes

Reviewer #3: None

PLOS authors have the option to publish the peer review history of their article (what does this mean?). If published, this will include your full peer review and any attached files.

Reviewer #1: No

Reviewer #2: No

Reviewer #3: No
---

## [Decision Letter · Decision Letter 1]

8 Sep 2022

Dear Dr Wen,

Thank you very much for submitting your manuscript "Rapid detection and recognition of whole brain activity in a freely behaving Caenorhabditis elegans" for consideration at PLOS Computational Biology. As with all papers reviewed by the journal, your manuscript was reviewed by members of the editorial board and by several independent reviewers. The reviewers appreciated the attention to an important topic. Based on the reviews, we are likely to accept this manuscript for publication, providing that you modify the manuscript according to the review recommendations. Please be sure to address all of the remaining items identified by the reviewers.

Sincerely,

Blake A Richards

Academic Editor

PLOS Computational Biology

Daniele Marinazzo

Section Editor

PLOS Computational Biology

[LINK]

Please be sure to address all of the remaining items identified by the reviewers.

Reviewer's Responses to Questions

**Comments to the Authors:**

Reviewer #1: The authors have addressed all my comments and the new manuscript is excellent. I'm excited to see the paper out!

One very minor correction: Table 2, "NeuroPAL Yu" row, the strain should be "AML32" not "AML320"

Reviewer #2: I believe that the authors have expertly addressed all the comments that I and the other reviewers raised. The paper seems, to me, clearly written; it contains both promising results and thoughtful commentary about challenges; it describes how this work relates to and advances upon the state of the art; and it contains a variety of attractive and informative figures that document all main claims.

Among changes, the analysis of the NeRVE dataset is particularly interesting, and provides a cross-experimenter validation of method that is very important but often missing in work such as this.

As I was reading the otherwise excellent paper, I found a small number of typos and/or instances of awkward wording. None of these greatly impair understanding of the scientific content, but I point them out here anyway:

(1) Line 6--"dynamic brain" should be "brain dynamics"

(2) Line 112--"filter slightly bigger" should be "filter be slightly bigger"

(3) Line 125--"detecting" should be "detect"

(4) Lines 301 & 302--"It is the consistency of a number across imaging volumes within an animal that allows the extraction" is grammatically correct but slightly awkward; the same point might be conveyed more comfortably with "Because it is consistent across imaging volumes within an animal, digital ID allows the extraction"

(5) Line 306--"have correspondence" might be clearer as "have a fixed correspondence across animals".

(6) Line 325--"thus have a" should be "thus has a"

(7) Line 431--"Noticeably" should probably be "Notably". (The clusters are noticeable, but "Notably" is the usual way to phrase such things.)

(8) Line 439--"at times when" should be "including a few cases where" (or something like that--it is important to point out that these are rare cases, which "at times when" doesn't do, in addition to suggesting that fluctuation is limited to the cases where it drops below 60%, which isn't really the case...the other fluctuations are just smaller).

(9) Line 442--"neurons, precluding an" should be "neurons. This precluded" (to help break up the overly-long sort-of-run-on sentence).

That is all!

Reviewer #3: The authors should be commended for making very useful additions including adding more rigorous quantification of performance and adding comparisons against other existing datasets. A few points remain that are still confusing in the manuscript’s current form (even though they were explained clearly in the most recent response to reviewers). If these straightforward clarifications are addressed in the manuscript, then I believe the work will be a valuable addition to the community. I recommend acceptance pending the following specific clarifications.

Important clarifications needed in the main text:

Most importantly, the language about “within-animal” vs “across-animal” and “digital ID”, vs “cell ID” remains confusing despite the authors efforts to clarify in the text. For example, when I look at Table 4 I am inclined to think that “across-animal” tracking means tracking cell IDs across animals. But instead it refers to tracking digital neuron IDs within an animal, just using *training* from across animals. Please further clarify to the reader so this is less confusing. For example, language such as “within-animal tracking, via across-animal training” seems more accurate and clear.

Similarly the definition of “digital ID” and “cell ID” (lines 298) and a mention that CeNDeR is only for “digital ID”s (line 471) should be made very early on in the paper and not a dozen pages in, so that a reader can quickly and accurately understand the scope of this work. For example around line 51 the authors should say something to the effective of “matching cell-type identities across animals is beyond the scope of the present work”.

Also in the author summary, the word “recognizes” in “detecting and recognizing most of the neurons” also seems to incorrectly convey give the impression that the algorithm finds cell IDs as opposed to digital neuron IDs. Please clarify in these areas too.

Additionally, it it is important to explicitly remind the reader in the Discussion section about the non-trivial amount of user input required (30 frames) as a current challenge of the approach.

Minor:

-I recommend plotting 8C and 8E on a single plot for easier comparison

-I think Figure 9CD would be more useful if it was combined so that the two algorithms could be compared on the same dataset

**Have the authors made all data and (if applicable) computational code underlying the findings in their manuscript fully available?**

Reviewer #1: Yes

Reviewer #2: Yes

Reviewer #3: Yes

PLOS authors have the option to publish the peer review history of their article (what does this mean?). If published, this will include your full peer review and any attached files.

Reviewer #1: No

Reviewer #2: No

Reviewer #3: No

Figure Files:

Data Requirements:

Reproducibility:

References:

---

## [Editor Report · Decision Letter 2]

22 Sep 2022

Dear Dr Wen,

We are pleased to inform you that your manuscript 'Rapid detection and recognition of whole brain activity in a freely behaving Caenorhabditis elegans' has been provisionally accepted for publication in PLOS Computational Biology.

Best regards,

Blake A Richards

Academic Editor

PLOS Computational Biology

Daniele Marinazzo

Section Editor

PLOS Computational Biology

---

## [Editor Report · Acceptance letter]

5 Oct 2022

PCOMPBIOL-D-22-00186R2 

Rapid detection and recognition of whole brain activity in a freely behaving *Caenorhabditis elegans*

Dear Dr Wen,

I am pleased to inform you that your manuscript has been formally accepted for publication in PLOS Computational Biology. Your manuscript is now with our production department and you will be notified of the publication date in due course.

With kind regards,

Agnes Pap
